# Label-free Node Classification on Graphs with Large Language Models (LLMs)

**Zhikai Chen**[1], **Haitao Mao**[1], **Hongzhi Wen**[1], **Haoyu Han**[1], **Wei Jin**[2],
**Haiyang Zhang**[3], **Hui Liu**[1], **Jiliang Tang**[1]
[1]Michigan State University   [2]Emory University   [3]Amazon.com
{chenzh85, haitaoma, wenhongz, hanhaoy1, liuhui7, tangjili}@msu.edu,
wei.jin@emory.edu, hhaiz@amazon.com

## Abstract

In recent years, there have been remarkable advancements in node classification achieved by Graph Neural Networks (GNNs). However, they necessitate abundant high-quality labels to ensure promising performance. In contrast, Large Language Models (LLMs) exhibit impressive zero-shot proficiency on text-attributed graphs. Yet, they face challenges in efficiently processing structural data and suffer from high inference costs. In light of these observations, this work introduces a label-free node classification on graphs with LLMs pipeline, LLM-GNN. It amalgamates the strengths of both GNNs and LLMs while mitigating their limitations. Specifically, LLMs are leveraged to annotate a small portion of nodes and then GNNs are trained on LLMs' annotations to make predictions for the remaining large portion of nodes. The implementation of LLM-GNN faces a unique challenge: how can we actively select nodes for LLMs to annotate and consequently enhance the GNN training? How can we leverage LLMs to obtain annotations of high quality, representativeness, and diversity, thereby enhancing GNN performance with less cost? To tackle this challenge, we develop an annotation quality heuristic and leverage the confidence scores derived from LLMs to advanced node selection. Comprehensive experimental results validate the effectiveness of LLM-GNN on text-attributed graphs from various domains. In particular, LLM-GNN can achieve an accuracy of 74.9% on a vast-scale dataset Ogbn-products with a cost less than 1 dollar. Our code is available from `https://github.com/CurryTang/LLMGNN`.

## 1 Introduction

Graphs are prevalent across multiple disciplines with diverse applications (Ma & Tang, 2021). A graph is composed of nodes and edges, and nodes often with certain attributes, especially text attributes, representing properties of nodes. For example, in the Ogbn-products dataset (Hu et al., 2020b), each node represents a product, and its corresponding textual description corresponds to the node's attribute. Node classification is a critical task for graphs which aims to assign labels to unlabeled nodes based on a part of labeled nodes, node attributes, and graph structures. In recent years, Graph Neural Networks (GNNs) have achieved superior performance in node classification (Kipf & Welling, 2016; Hamilton et al., 2017; Veličković et al., 2017). Despite the effectiveness of GNNs, they always assume the ready availability of ground truth labels as a prerequire. Particularly, such assumption often neglects the pivotal challenge of procuring high-quality labels for graph-structured data: (1) given the diverse and complex nature of graph-structured data, human labeling is inherently hard; (2) given the sheer scale of real-world graphs, such as Ogbn-products (Hu et al., 2020b) with millions of nodes, the process of annotating a significant portion of the nodes becomes both time-consuming and resource-intensive.

Compared to GNNs which require adequate high-quality labels, Large Language Models (LLMs) with massive knowledge have showcased impressive zero-shot and few-shot capabilities, especially for the node classification task on text-attributed graphs (TAGs) (Guo et al., 2023; Chen et al., 2023; He et al., 2023a). Such evidence suggests that LLMs can achieve promising performance with-

out the requirement for any labeled data. However, unlike GNNs, LLMs cannot naturally capture and understand informative graph structural patterns (Wang et al., 2023a). Moreover, LLMs can not be well-tuned since they can only utilize limited labels due to the limitation of input context length (Dong et al., 2022). Thus, though LLMs can achieve promising performance in zero-shot or few-shot scenarios, there may still be a performance gap between LLMs and GNNs trained with abundant labeled nodes (Chen et al., 2023). Furthermore, the prediction cost of LLMs is much higher than that of GNNs, making it less scalable for large datasets such as OGBN-ARXIV and OGBN-PRODUCTS (Hu et al., 2020b).

In summary, we make two primary observations: (1) Given adequate annotations with high quality, GNNs excel in utilizing graph structures to provide predictions both efficiently and effectively. Nonetheless, limitations can be found when adequate high-quality annotations are absent. (2) In contrast, LLMs can achieve satisfying performance without high-quality annotations while being costly. Considering these insights, it becomes evident that GNNs and LLMs possess complementary strengths. This leads us to an intriguing question: Can we harness the strengths of both while addressing their inherent weaknesses?

In this paper, we provide an affirmative answer to the above question by investigating the potential of harnessing the zero-shot learning capabilities of LLMs to alleviate the substantial training data demands of GNNs, a scenario we refer to as *label-free node classification*. Notably, unlike the common assumption that ground truth labels are always available, noisy labels can be found when annotations are generated from LLMs. We thus confront a unique challenge: How can we ensure the high quality of the annotation without sacrifice diversity and representativeness? On one hand, we are required to consider the design of appropriate prompts to enable LLMs to produce more accurate annotations. On the other hand, we need to strategically choose a set of training nodes that not only possess high-quality annotations but also exhibit informativeness and representativeness, as prior research has shown a correlation between these attributes and the performance of the trained model (Huang et al., 2010).

To overcome these challenges, we propose a **label-free node classification on graphs with LLMs** pipeline, LLM-GNN. Different from traditional graph active node selection (Wu et al., 2019; Cai et al., 2017), LLM-GNN considers the node annotation difficulty by LLMs to actively select nodes. Then, it utilizes LLMs to generate confidence-aware annotations and leverages the confidence score to further refine the quality of annotations as post-filtering. By seamlessly blending annotation quality with active selection, LLM-GNN achieves impressive results at a minimal cost, eliminating the necessity for ground truth labels. Our main contributions can be summarized as follows:

1. We introduce a new label-free pipeline LLM-GNN to leverage LLMs for annotation, providing training signals on GNN for further prediction.
2. We adopt LLMs to generate annotations with calibrated confidence, and introduce difficulty-aware active selection with post filtering to get training nodes with a proper trade-off between annotation quality and traditional graph active selection criteria.
3. On the massive-scale OGBN-PRODUCTS dataset, LLM-GNN can achieve **74.9% accuracy** without the need for human annotations. This performance is comparable to manually annotating 400 randomly selected nodes, while the cost of the annotation process via LLMs is **under 1 dollar**.

## 2 PRELIMINARIES

In this section, we introduce text-attributed graphs and notation utilized in our study. We then review two primary pipelines on node classification. The first pipeline is the default node classification pipeline to evaluate the performance of GNNs (Kipf & Welling, 2016), while it totally ignores the data selection process. The second pipeline further emphasizes the node selection process, trying to identify the most informative nodes as training sets to maximize the model performance within a given budget.

Our study focuses on *Text-Attributed Graph (TAG)*, represented as $\mathcal{G}_T = (\mathcal{V}, \mathbf{A}, \mathbf{T}, \mathbf{X})$. $\mathcal{V} = \{v_1, \cdots, v_n\}$ is the set of $n$ nodes paired with raw attributes $\mathbf{T} = \{\mathbf{t}_1, \mathbf{t}_2, \ldots, \mathbf{t}_n\}$. Each text attributes can then be encoded as sentence embedding $\mathbf{X} = \{\boldsymbol{x_1}, \boldsymbol{x_2}, \ldots, \boldsymbol{x_n}\}$ with the help of SentenceBERT (Reimers & Gurevych, 2019). The adjacency matrix $\mathbf{A} \in \{0, 1\}^{n \times n}$ represents graph connectivity where $\mathbf{A}[i, j] = 1$ indicates an edge between nodes $i$ and $j$. Although our study puts more emphasis on TAGs, it has the potential to be extended to more types of graphs through methods like Liu et al. (2023) and Zhao et al. (2023).

**Traditional GNN-based node classification pipeline.** assumes a fixed training set with ground truth labels $y_{\mathcal{V}_{train}}$ for the training set $\mathcal{V}_{train}$. The GNN is trained on those graph truth labels. The well-trained GNN predicts labels of the rest unlabeled nodes $\mathcal{V} \setminus \mathcal{V}_{train}$ in the test stage.

**Traditional graph active learning-based node classification.** aims to select a group of nodes $\mathcal{V}_{act} = \mathcal{S}(\mathbf{A}, \mathbf{X})$ from the pool $\mathcal{V}$ so that the performance of GNN models trained on those graphs with labels $y_{\mathcal{V}_{act}}$ can be maximized.

**Limitations of the current pipelines.** Both pipelines above assume they can always obtain ground truth labels (Zhang et al., 2021c; Wu et al., 2019) while overlooking the intricacies of the annotation process. Nonetheless, annotations can be both expensive and error-prone in practice, even for seemingly straightforward tasks (Wei et al., 2021). For example, the accuracy of human annotations for the CIFAR-10 dataset is approximately 82%, which only involves the categorization of daily objects. Annotation on graphs meets its unique challenge. Recent evidence (Zhu et al., 2021a) shows that the human annotation on graphs is easily biased, focusing nodes sharing some characteristics within a small subgraph. Moreover, it can be even harder when taking graph active learning into consideration, the improved annotation diversity inevitably increase the difficulty of ensuring the annotation quality. For instance, it is much easier to annotate focusing on a few small communities than annotate across all the communities in a social network. Considering these limitations of existing pipelines, a pertinent question arises: *Can we design a pipeline that can leverage LLMs to automatically generate high-quality annotations and utilize them to train a GNN model with promising node classification performance?*

## 3 METHOD

To overcome the limitations of current pipelines for node classifications, we propose a new pipeline **Label-free Node Classification on Graphs with LLMs**, short for *LLM-GNN*. It (1) adopts LLMs that demonstrate promising zero-shot performance on various node classification datasets (Chen et al., 2023; He et al., 2023a) as the annotators; and (2) introduces the (difficulty-aware) active selection and optional filtering strategy to get training nodes with high annotation quality, representativeness, and diversity simultaneously.

### 3.1 AN OVERVIEW OF LLM-GNN

The proposed LLM-GNN pipeline is designed with four flexible components as shown in Figure 1: *(difficulty-aware) active node selection, (confidence-aware annotations), optional post-filtering*, and *GNN model training and prediction*. Compared with the original pipelines with ground truth label, the annotation quality of LLM provides a unique new challenge. **(1)** The active node selection phase is to find a candidate node set for LLM annotation. Despite only considering the diversity and representativeness (Zhang et al., 2021c) as the original baseline, we pay additional attention to the influence on annotation quality. Specifically, we incorporate a difficulty-aware heuristic that correlates the annotation quality with the feature density. **(2)** With the selected node set, we then utilize the strong zero-shot ability of LLMs to annotate those nodes with confidence-aware prompts. The confidence score associated with annotations is essen-

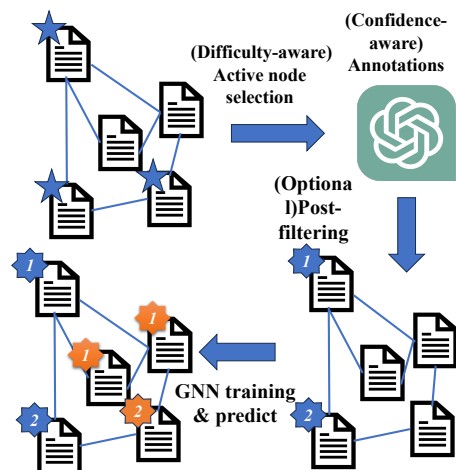

Figure 1: Our proposed pipeline *LLM-GNN* for label-free node classification on graphs. It is designed with four components: (1)*(difficulty-aware) active node selection* to select suitable nodes which are easy to annotate. (2) *(confidence-aware annotations)* generate the annotation on the node set and confidence score reflecting the label quality. (3) *optional post-filtering* removes low quality annotation via confidence score (4) *GNN model training and prediction*.

tial, as LLM annotations (Chen et al., 2023), akin to human annotations, can exhibit a certain degree of label noise. This confidence score can help to identify the annotation quality and help us filter high-quality labels from noisy ones. **(3)** The optional post-filtering stage is a unique step in our pipeline, which aims to remove low-quality annotations. Building upon the annotation confidence, we further refine the quality of annotations with LLMs' confidence scores and remove those nodes

with lower confidence from the previously selected set and **(4)** With the filtered high-quality annotation set, we then train GNN models on selected nodes and their annotations. Ultimately, the well-trained GNN model is then utilized to perform predictions. It should be noted that the framework we propose is very flexible, with different designs possible for each part. For example, for the part of active node selection, we can use conventional active learning methods combined with post-filtering to improve the overall quality of the labeling. We then detail each component.

### 3.2 DIFFICULTY-AWARE ACTIVE NODE SELECTION

Node selection aims to select a node candidate set, which will be annotated by LLMs, and then learned on GNN. Notably, the selected node set is generally small to ensure a controllable money budget. Unlike traditional graph active learning which mainly takes diversity and representativeness into consideration, label quality should also be included since LLM can produce noisy labels with large variances across different groups of nodes.

In the difficulty-aware active selection stage, we have no knowledge of how LLMs would respond to those nodes. Consequently, we are required to identify some heuristics building connections to the difficulty of annotating different nodes. A preliminary investigation of LLMs' annotations brings us inspiration on how to infer the annotation difficulty through node features, where we find that the accuracy of annotations generated by LLMs is closely related to the clustering density of nodes.

To demonstrate this correlation, we employ k-means clustering on the original feature space, setting the number of clusters equal to the distinct class count. 1000 nodes are sampled from the whole dataset and then annotated by LLMs. They are subsequently sorted and divided into ten equally sized groups based on their distance to the nearest cluster center. As shown in Figure 2, we observe a consistent pattern: *nodes closer to cluster centers typically exhibit better annotation quality, which indicates lower annotation difficulty*. Full results are included in Appendix F.2. We then adopt this distance as a heuristic to approximate the annotation reliability. Since the cluster number equals the distinct class count, we denote this heuristic as C-Density, calculcated as C-Density$(v_i) = \frac{1}{1+\|x_{v_i}-x_{\text{CC}_{v_i}}\|}$, where for any node $v_i$ and its closest cluster center $\text{CC}_{v_i}$, and $x_{v_i}$ represents the feature of node $v_i$. We demonstrate the effectiveness of this method through theoretical explanation in Appendix K.

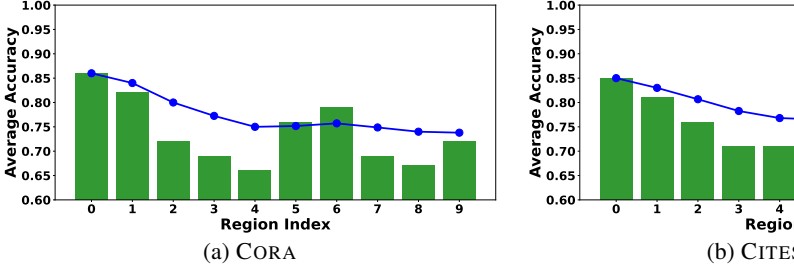

|       | (a) CORA | (b) CITESEER |
|-------|----------|--------------|

Figure 2: The annotation accuracy by LLMs vs. the distance to the nearest clustering center. The bars represent the average accuracy within each selected group, while the blue line indicates the cumulative average accuracy. At group $i$, the blue line denotes the average accuracy of all nodes in the preceding $i$ groups.

We then incorporate this annotation difficulty heuristic in traditional active selection. For traditional active node selection, we select $B$ nodes from the unlabeled pools with top scores $f_{\text{act}}(v_i)$, where $f_{\text{act}}()$ is a score function (we defer the detailed introduction of $f_{\text{act}}()$ to Appendix A). To benefit the performance of the trained models, the selected nodes should have a trade-off between annotation difficulty and traditional active selection criteria ( e.g., representativeness (Cai et al., 2017) and diversity (Zhang et al., 2021c)). In traditional graph active learning, selection criteria can usually be denoted as a score function, such as PageRank centrality $f_{\text{pg}}(v_i)$ for measuring the structural diversity. Then, one feasible way to integrate difficulty heuristic into traditional graph active learning is ranking aggregation. Compared to directly combining several scores via summation or multiplication, ranking aggregation is more robust to scale differences since it is scale-invariant and considers only the relative ordering of items. Considering the original score function for graph active learning as $f_{\text{act}}(v_i)$, we denote $r_{f_{\text{act}}}(v_i)$ as the high-to-low ranking percentage. Then, we incorporate difficulty heuristic by first transforming C-Density$(v_i)$ into a rank $r_{\text{C-Density}}(v_i)$. Then we combine these two scores: $f_{\text{DA-act}}(v_i) = \alpha_0 \times r_{f_{\text{act}}}(v_i) + \alpha_1 \times r_{\text{C-Density}}(v_i)$. "DA" stands for difficulty aware. Hyperparameters $\alpha_0$ and $\alpha_1$ are introduced to balance annotation difficulty and traditional graph active

learning criteria such as representativeness, and diversity. Finally, nodes $v_i$ with larger $f_{\text{DA-act}}(v_i)$ are selected for LLMs to annotate, which is denoted as $\mathcal{V}_{\text{anno}}$.

Table 1: Accuracy of annotations. Yellow denotes the best and Green denotes the second best result. The cost is determined by comparing the token consumption to that of zero-shot prompts.

| Prompt Strategy | CORA | | OGBN-PRODUCTS | | WIKICS | |
|---|---|---|---|---|---|---|
| | Acc (%) | Cost | Acc (%) | Cost | Acc (%) | Cost |
| Vanilla (zero-shot) | 68.33 ± 6.55 | 1 | 75.33 ± 4.99 | 1 | 68.33 ± 1.89 | 1 |
| Vanilla (one-shot) | 69.67 ± 7.72 | 2.2 | 78.67 ± 4.50 | 1.8 | 72.00 ± 3.56 | 2.4 |
| TopK (zero-shot) | 68.00 ± 6.38 | 1.1 | 74.00 ± 5.10 | 1.2 | 72.00 ± 2.16 | 1.1 |
| Most Voting (zero-shot) | 68.00 ± 7.35 | 1.1 | 75.33 ± 4.99 | 1.1 | 69.00 ± 2.16 | 1.1 |
| Hybrid (zero-shot) | 67.33 ± 6.80 | 1.5 | 73.67 ± 5.25 | 1.4 | 71.00 ± 2.83 | 1.4 |
| Hybrid (one-shot) | 70.33 ± 6.24 | 2.9 | 75.67 ± 6.13 | 2.3 | 73.67 ± 2.62 | 2.9 |

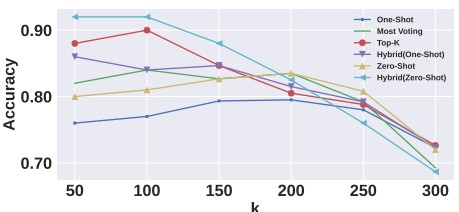

Figure 3: An illustration of the relation between accuracy and confidence on WIKICS.

## 3.3 CONFIDENCE-AWARE ANNOTATIONS

After obtaining the candidate set $\mathcal{V}_{\text{anno}}$ through active selection, we use LLMs to generate annotations for nodes in the set. Despite we select nodes easily to annotate with difficulty-aware active node selection, we are not aware of how the LLM responds to the nodes at that stage. It leads to the potential of remaining low-quality nodes. To figure out the high-quality annotations, we need some guidance on their reliability, such as the confidence scores. Inspired by recent literature on generating calibrated confidence from LLMs (Xiong et al., 2023; Tian et al., 2023; Wang et al., 2022), we investigate the following strategies: (1) directly asking for confidence (Tian et al., 2023), denoted as "Vanilla (zero-shot)"; (2) reasoning-based prompts to generate annotations, including chain-of-thought and multi-step reasoning (Wei et al., 2022; Xiong et al., 2023); (3) TopK prompt, which asks LLMs to generate the top $K$ possible answers and select the most probable one as the answer (Xiong et al., 2023); (4) Consistency-based prompt (Wang et al., 2022), which queries LLMs multiple times and selects the most common output as the answer, denoted as "Most voting"; (5) Hybrid prompt (Wang et al., 2023a), which combines both TopK prompt and consistency-based prompt. In addition to the prompt strategy, few-shot samples have also been demonstrated critical to the performance of LLMs (Chen et al., 2023). We thus also investigate incorporating few-shot samples into prompts. In this work, we try 1-shot sample to avoid time and money cost. Detailed descriptions and full prompt examples are shown in Appendix D.

Then, we do a comparative study to identify the effective prompts in terms of accuracy, calibration of confidence, and costs. (1) For accuracy and cost evaluation, we adopt popular node classification benchmarks CORA, OGBN-PRODUCTS, and WIKICS. We randomly sample 100 nodes from each dataset and repeat with three different seeds to reduce sampling bias, and then we compare the generated annotations with the ground truth labels offered by these datasets. The cost is estimated by the number of tokens in input prompts and output contents. (2) It is important to note that the role of confidence is to assist us in identifying label reliability. Therefore, To validate the quality of the confidence produced by LLMs is to examine how the confidence can reflect the quality of the corresponding annotation. Therefore, we check how the annotation accuracy changes with the confidence. Specifically, we randomly select 300 nodes and sort them in descending order based on their confidence. Subsequently, we calculate the annotation accuracy for the top $k$ nodes where $K$ is varied in $\{50, 100, 150, 200, 250, 300\}$. For each $K$, a higher accuracy indicates a better quality of the generated confidence. Empirically, we find that reasoning-based prompts will generate outputs that don't follow the format requirement, and greatly increase the query time and costs. Therefore, we don't further consider them in this work. For other prompts, we find that for a small portion of inputs, the outputs of LLMs do not follow the format requirements (for example, outputting an annotation out of the valid label names). For those invalid outputs, we design a self-correction prompt and set a larger temperature to review previous outputs and regenerate annotations.

The evaluation of performance and cost is shown in Table 1, and the evaluation of confidence is shown in Figure 3. The full results are included in Appendix E. From the experimental results, we make the following observations: **First**, LLMs present promising zero-shot prediction performance on all datasets, which suggests that LLMs are potentially good annotators. **Second**, compared to zero-shot prompts, prompts with few-shot demonstrations could slightly increase performance with double costs. **Third**, zero-shot hybrid strategies present the most effective approach to extract high-quality annotations since the confidence can greatly indicate the quality of annotation. We thus adopt

zero-shot hybrid prompt in the following studies and leave the evaluation of other prompts as one future work.

### 3.4 POST-FILTERING

After achieving annotations together with the confidence scores, we may further refine the set of annotated nodes since we have access to confidence scores generated by LLMs, which can be used to filter high-quality labels. However, directly filtering out low-confidence nodes may result in a label distribution shift and degrade the diversity of the selected nodes, which will degrade the performance of subsequently trained models. Unlike traditional graph active learning methods which try to model diversity in the selection stage with criteria such as feature dissimilarity (Ren et al., 2022), in the post-filtering stage, label distribution is readily available. As a result, we can directly consider the label diversity of selected nodes. To measure the change of diversity, we propose a simple score function *change of entropy (COE)* to measure the entropy change of labels when removing a node from the selected set. Assuming that the current selected set of nodes is $\mathcal{V}_{\text{sel}}$, then COE can be computed as: $\text{COE}(v_i) = \boldsymbol{H}(\tilde{y}_{\mathcal{V}_{\text{sel}} - \{v_i\}}) - \boldsymbol{H}(\tilde{y}_{\mathcal{V}_{\text{sel}}})$ where $\boldsymbol{H}()$ is the Shannon entropy function (Shannon, 1948), and $\tilde{y}$ denotes the annotations generated by LLMs. It should be noted that the value of COE may possibly be positive or negative, and a small $\text{COE}(v_i)$ value indicates that removing this node could adversely affect the diversity of the selected set, potentially compromising the performance of trained models. When a node is removed from the selected set, the entropy adjusts accordingly, necessitating a re-computation of COE. However, it introduces negligible computation overhead since the size of the selected set $\mathcal{V}_{\text{anno}}$ is usually much smaller than the whole dataset. COE can be further combined with confidence $f_{\text{conf}}(v_i)$ to balance diversity and annotation quality, in a ranking aggregation manner. It should be noted that $r_{\text{C-Density}}(v_i)$ is also available in the post filtering phase. So, the final filtering score function $f_{\text{filter}}$ can be stated as: $f_{\text{filter}}(v_i) = \beta_0 \times r_{f_{\text{conf}}(v_i)} + \beta_1 \times r_{\text{COE}(v_i)} + \beta_2 \times r_{\text{C-Density}}(v_i)$. Hyper-parameters $\beta_0$, $\beta_1$, and $\beta_2$ are introduced to balance label diversity and annotation quality. $r_{f_{\text{conf}}}$ is the high-to-low ranking percentage of the confidence score $f_{\text{conf}}$. To conduct post-filtering, each time we remove the node with the smallest $f_{\text{filter}}$ value until a pre-defined maximal number is reached.

### 3.5 GNN TRAINING AND PREDICTION

After obtaining the training labels, we further train a GNN. Our framework supports a variety of GNNs, and we select GCN, the most popular model, as our primary subject of study. Additionally, another critical component during the training process is the loss function. Traditional GNN-based pipelines mainly adopt cross-entropy loss; however, due to the noisy labels generated by the LLMs, we may also utilize a weighted cross-entropy loss in this part. Specifically, we can use the confidence scores from the previous section as the corresponding weights.

## 4 EXPERIMENT

In this section, we present experiments to evaluate the performance of our proposed pipeline *LLM-GNN*. We begin by detailing the experimental settings. Next, we investigate the following research questions: **RQ1.** How do active selection, post-filtering, and loss function affect the performance of *LLM-GNN*? How to come up with an effective pipeline implementation? **RQ2.** How does the performance and cost of *LLM-GNN* compare to other label-free node classification methods? **RQ3.** How do different budgets affect the performance of the pipelines? **RQ4.** How do LLMs' annotations compare to ground truth labels?

### 4.1 EXPERIMENTAL SETTINGS

In this paper, we adopt the following TAG datasets widely adopted for node classification: CORA (McCallum et al., 2000), CITESEER (Giles et al., 1998), PUBMED (Sen et al., 2008), OGBN-ARXIV, OGBN-PRODUCTS (Hu et al., 2020b), and WIKICS (Mernyei & Cangea, 2020). Statistics and descriptions of these datasets are in Appendix C.

In terms of the settings for each component in the pipeline, we adopt gpt-3.5-turbo-0613 [1] to generate annotations. In terms of the prompt strategy for generating annotations, we choose the "zero-shot hybrid strategy" considering its effectiveness in generating calibrated confidence. We leave the evaluation of other prompts as future works considering the massive costs. For the budget of the active selection, we refer to the popular semi-supervised learning setting for node classifications (Yang et al., 2016) and set the budget equal to 20 multiplied by the number of classes. For GNNs, we adopt one of the most popular models GCN (Kipf & Welling, 2016). *The major goal of this evaluation is*

---
[1] https://platform.openai.com/docs/models/gpt-3-5

Table 2: Impact of different active selection strategies. We show the top three performance in each dataset with pink, green, and yellow, respectively. To compare with traditional graph active selections, we underline their combinations with our selection strategies if the combinations outperform their corresponding traditional graph active selection. OOT means that this method can not scale to large-scale graphs because of long execution time.

| | CORA | CITESEER | PUBMED | WIKICS | OGBN-ARXIV | OGBN-PRODUCTS |
|---|---|---|---|---|---|---|
| Random | 70.48± 0.73 | 65.11 ± 1.12 | 72.98 ± 2.15 | 60.69 ± 1.73 | 64.59 ± 0.16 | 70.40 ± 0.60 |
| Random-W | 71.77± 0.75 | 65.92 ± 1.05 | 73.92 ± 1.75 | 61.42± 1.54 | 64.95 ± 0.19 | 71.96± 0.59 |
| C-Density | 42.22 ± 1.59 | 66.44 ± 0.34 | 74.43 ± 0.28 | 57.77 ± 0.85 | 44.08 ± 0.39 | 8.29 ± 0.00 |
| PS-Random-W | 72.38 ± 0.72 | 67.18 ± 0.92 | 73.31 ± 1.65 | 62.60 ± 0.94 | 65.22 ± 0.15 | 71.62± 0.54 |
| Density | 72.40 ± 0.35 | 61.06 ± 0.95 | 74.43 ± 0.28 | 64.96 ± 0.53 | 51.77 ± 0.24 | 20.22 ± 0.11 |
| Density-W | 72.39± 0.34 | 59.88 ± 0.97 | 73.00 ± 0.19 | 63.80 ± 0.69 | 51.03 ± 0.27 | 20.97 ± 0.15 |
| DA-Density | 70.73 ± 0.32 | 62.92 ±1.05 | 74.43 ± 0.28 | 63.08 ± 0.45 | 51.33 ± 0.29 | 8.50 ± 0.32 |
| PS-Density-W | 74.61 ± 0.13 | 61.00 ± 0.55 | 74.50 ± 0.23 | 65.57 ± 0.45 | 51.73 ± 0.29 | 19.15 ± 0.18 |
| DA-Density-W | 67.29 ± 0.96 | 62.98 ± 0.77 | 73.39 ± 0.35 | 63.26 ± 0.62 | 51.36 ± 0.39 | 8.52 ± 0.11 |
| AGE | 69.15 ± 0.38 | 54.25 ± 0.31 | 74.55 ± 0.54 | 55.51 ± 0.12 | 46.68 ± 0.30 | 65.63 ± 0.15 |
| AGE-W | 69.70 ± 0.45 | 57.60 ± 0.35 | 64.30 ± 0.49 | 55.15 ± 0.14 | 47.84± 0.35 | 64.92 ± 0.19 |
| DA-AGE | 74.38 ± 0.24 | 59.92 ± 0.42 | 74.20 ± 0.51 | 59.39 ± 0.21 | 48.21 ± 0.35 | 60.03 ± 0.11 |
| PS-AGE-W | 72.61 ± 0.39 | 57.44 ± 0.49 | 64.00 ± 0.44 | 56.13 ± 0.11 | 47.12 ± 0.39 | 68.62 ± 0.15 |
| DA-AGE-W | 74.96 ± 0.22 | 58.41 ± 0.45 | 65.85 ± 0.67 | 59.19 ± 0.24 | 47.79± 0.32 | 59.95 ± 0.23 |
| RIM | 69.86 ± 0.38 | 63.44 ± 0.42 | 76.22 ± 0.16 | 66.72± 0.16 | OOT | OOT |
| DA-RIM | 73.99 ± 0.44 | 60.33 ± 0.40 | 79.17 ± 0.11 | 67.82± 0.32 | OOT | OOT |
| PS-RIM-W | 73.19 ± 0.45 | 62.85 ± 0.49 | 74.52 ± 0.19 | 69.84 ± 0.19 | OOT | OOT |
| DA-RIM-W | 74.73 ± 0.41 | 60.80 ± 0.57 | 77.94 ± 0.24 | 68.22 ± 0.25 | OOT | OOT |
| GraphPart | 68.57 ± 2.18 | 66.59 ± 1.34 | 77.50 ± 1.23 | 67.28 ± 0.87 | OOT | OOT |
| GraphPart-W | 69.90 ± 2.03 | 68.20 ± 1.42 | 78.91 ± 1.04 | 68.43 ± 0.92 | OOT | OOT |
| DA-GraphPart | 69.35 ± 1.92 | 69.37 ± 1.27 | 79.49 ± 0.85 | 68.72 ± 1.01 | OOT | OOT |
| PS-GraphPart-W | 69.92± 1.75 | 69.06 ± 1.19 | 78.84 ± 1.05 | 66.90 ± 1.05 | OOT | OOT |
| DA-GraphPart-W | 68.61 ± 1.32 | 68.82 ± 1.17 | 79.89 ± 0.79 | 67.13 ± 1.23 | OOT | OOT |
| FeatProp | 72.82 ± 0.08 | 66.61 ± 0.55 | 73.90 ± 0.15 | 64.08 ± 0.12 | 66.06 ± 0.07 | 74.04 ± 0.15 |
| FeatProp-W | 73.56 ± 0.13 | 68.04 ± 0.69 | 76.90 ± 0.19 | 63.80 ± 0.21 | 66.32 ± 0.15 | 74.32 ± 0.14 |
| PS-FeatProp | 75.54 ± 0.34 | 69.06 ± 0.32 | 74.98 ± 0.35 | 66.09 ± 0.35 | 66.14 ± 0.27 | 74.91 ± 0.17 |
| PS-FeatProp-W | 76.23 ± 0.07 | 68.64 ± 0.71 | 78.84 ± 1.05 | 64.72 ± 0.19 | 65.84 ± 0.19 | 74.54 ± 0.24 |

*to show the potential of the LLM-GNN pipeline. Therefore, we do not tune the hyper-parameters in both difficulty-aware active selection and post-filtering but simply setting them with the same value.*

In terms of evaluation, we compare the generated prediction of GNNs with the ground truth labels offered in the original datasets and adopt accuracy as the metric. Similar to (Ma et al., 2022), we adopt a setting where there's no validation set, and models trained on selected nodes will be further tested based on the rest unlabeled nodes. All experiments will be repeated for 3 times with different seeds. For hyper-parameters of the experiment, we adopt a fixed setting commonly used by previous papers or benchmarks Kipf & Welling (2016); Hamilton et al. (2017); Hu et al. (2020b). One point that should be strengthened is the number of training epochs. Since there's no validation set and the labels are noisy, models may suffer from overfitting (Song et al., 2022). However, we find that most models work well across all datasets by setting a small fixed number of training epochs, such as 30 epochs for small and medium-scale datasets (CORA, CITESEER, PUBMED, and WIKICS), and 50 epochs for the rest large-scale datasets. This setting, which can be viewed as a simpler alternative to early stop trick (without validation set) (Bai et al., 2021) for training on noisy labels so that we can compare different methods more fairly and conveniently.

## 4.2 RQ1. IMPACT OF DIFFERENT ACTIVE SELECTION STRATEGIES

We conduct a comprehensive evaluation of different active selection strategies, which is the key component of our pipeline. Specifically, we examine how effectiveness of (1) difficulty-aware active node selection before LLM annotation (2) post-filtering after LLM annotation and how they combine with traditional active learning algorithms (3) loss functions. For selection strategies, we consider (1) Traditional graph active selection: Random selection, Density-based selection (Ma et al., 2022), GraphPart (Ma et al., 2022), FeatProp (Wu et al., 2019), Degree-based selection (Ma et al., 2022), Pagerank centrality-based selection (Ma et al., 2022), AGE (Cai et al., 2017), and RIM (Zhang et al., 2021b). (2) Difficulty-aware active node selection: C-Density-based selection, and traditional graph active selections combined with C-Density. To denote these selections, we add the prefix **"DA-"**.

For example, "DA-AGE" means combining the original AGE method with our proposed C-Density. (3) Post Filtering: Traditional graph active selection combined with confidence and COE-based selections, we add the prefix **"PS-"**. For FeatProp, as it selects candidate nodes directly using the K-Medoids algorithm (Wu et al., 2019), integrating it with difficulty-aware active selections is not feasible. For loss functions, we consider both cross entropy loss and weighted cross entropy loss, where we add a "-W" postfix for the latter. Detailed introductions of these methods are shown in Appendix A. The results for GCN are shown in Table 2. In terms of space limits, we move part of the results and more ablation studies to Appendix J.

From the experimental results, we make the following observations:

1. The proposed post-filtering strategy presents promising effectiveness. Combined with traditional graph active learning methods like GraphPart, RIM, and Featprop, it can consistently outperform. Combined with FeatProp, it can achieve both promising accuracy and better scalability.
2. Although C-Density-based selection can achieve superior annotation quality, merely using this metric will make the trained model achieve poor performance. To better understand this phenomenon, we check the labels of the selected nodes. We find that the problem lies in label imbalance brought by active selection. For example, we check the selected nodes for PUBMED, and find that all annotations belong to one class. We further find that tuning the number of clustering centers for C-Density can trade off between diversity and annotation quality, where a larger $K$ can mitigate the class imbalance problem. However, it proposes a challenge to find a proper $K$ for massive-scale datasets like OGBN-PRODUCTS, where weighted loss and post-filtering are more effective.
3. Comparing normal cross entropy loss to weighted cross entropy loss, weighted cross entropy loss further enhance the performance for most of the cases.
4. In a nutshell, we summarize the following empirical rules of thumbs: (1) Featprop-based methods can consistently achieve promising performance across different datasets efficiently; (2) Comparing DA and PS, DA costs less since we don't need LLMs to generate the confidence and we may use a simpler prompt. PS can usually get better performance. On large-scale datasets, PS usually get much better results.

### 4.3 **(RQ2.)** COMPARISON WITH OTHER LABEL-FREE NODE CLASSIFICATION METHODS

To demonstrate the effectiveness and novelty of our proposed pipeline, we further conduct a comparison with other label-free node classification pipelines, which include: (1) Zero-shot node classification method: SES, TAG-Z (Li & Hooi, 2023); (2) Zero-shot classification models for texts: BART-large-MNLI (Lewis et al., 2019); and (3) Directly using LLMs for predictions: LLMs-as-Predictors (Chen et al., 2023). Detailed introductions of these models can be found in Appendix A. We compare both performance and costs of these models, and the results are shown in Table 3.

Table 3: Comparison of label-free node classification methods. The cost is computed in dollars. The performance of methods with * are taken from Li & Hooi (2023). Notably, the time cost of LLMs is proportional to the expenses.

|  | OGBN-ARXIV | | OGBN-PRODUCTS | |
| --- | --- | --- | --- | --- |
| Methods | Acc | Cost | Acc | Cost |
| SES(*) | 13.08 | N/A | 6.67 | N/A |
| TAG-Z(*) | 37.08 | N/A | 47.08 | N/A |
| BART-large-MNLI | 13.2 | N/A | 28.8 | N/A |
| LLMs-as-Predictors | 73.33 | 79 | 75.33 | 1572 |
| LLM-GNN | 66.32 | 0.63 | 74.91 | 0.74 |

From the experimental results in the table, we can see that (1) our proposed pipeline LLM-GNN can significantly outperform SES, TAG-Z and BART-large-MNLI. (2) Despite LLMs-as-Predictors has better performance than LLM-GNN, its cost is much higher than LLM-GNN. For example, the cost of LLMs-as-Predictors in OGBN-PRODUCTS is $2,124\times$ that of LLM-GNN. Besides, the promising performance of LLMs-as-Predictors on OGBN-ARXIV may be an exception, relevant to the specific prompts leveraging the memorization of LLMs (Chen et al., 2023).

### 4.4 **(RQ3.)** HOW DO DIFFERENT BUDGETS AFFECT THE PERFORMANCE OF OUR PIPELINES?

We conduct a comprehensive evaluation on different budgets rather the fixed budget in previous experiments. It aims to examine how effective our algorithm is when confronting different real-world scenarios with different to meet different cost and performance requirements. Experiments are typically conducted on the CORA dataset by setting the budget as $\{35, 70, 105, 140, 175, 280, 560, 1,120\}$. We choose both random selections and those methods that perform well in Table 2. We can have the following observations from Figure 4. (1) with the increase in the budget, the performance tends to increase gradually. (2) unlike using ground truth, the performance growth

is relatively limited as the budget increases. It suggests that there exists a trade-off between the performance and the cost in the real-world scenario.

### 4.5 (RQ4.) CHARACTERISTICS OF LLMS' ANNOTATIONS

Although LLMs' annotations are noisy labels, we find that they are more benign than the ground truth labels injected with synthetic noise adopted in (Zhang et al., 2021b). Specifically, assuming that the quality of LLMs' annotations is $q\%$, we randomly select $(1 - q)\%$ ground truth labels and flip them into other classes uniformly to generate synthetic noisy labels. We then train GNN models on LLMs' annotations, synthetic noisy labels, and LLMs' annotations with all incorrect labels removed, respectively. The results are demonstrated in Figure 5, from which we observe that: LLMs' annotations present totally different training dynamics from synthetic noisy labels. The extent of over-fitting for LLMs' annotations is much less than that for synthetic noisy labels.

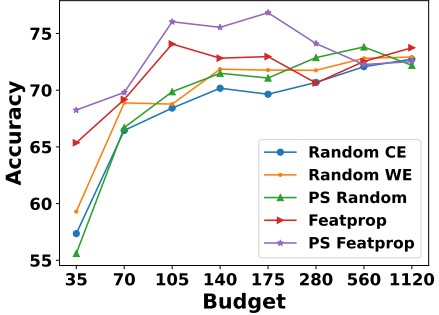
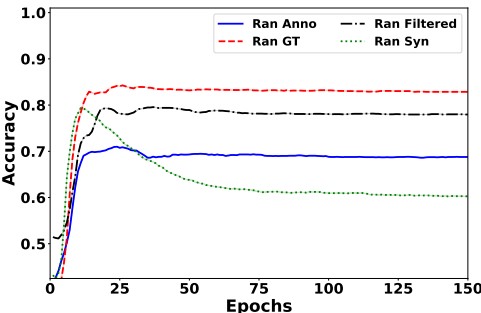

Figure 4: Investigation on how different budgets affect the performance of LLM-GNN. Methods achieving top performance in Table 2 and random selection-based methods are compared in the figure.CE and WE means normal cross entropy loss and weighted loss, respectively.

Figure 5: Comparisons among LLMs' annotations, ground truth labels, and synthetic noisy labels. "Ran" represents random selection, "GT" indicates the ground truth labels. "Filtered" means replacing all wrong annotations with ground truth labels.

## 5 RELATED WORKS

**Graph active learning.** Graph active learning (Cai et al., 2017) aims to maximize the test performance with nodes actively selected under a limited query budget. To achieve this goal, algorithms are developed to maximize the informativeness and diversity of the selected group of nodes (Zhang et al., 2021c). These algorithms are designed based on assumptions. In Ma et al. (2022), diversity is assumed to be related to the partition of nodes, and thus samples are actively selected from different communities. In Zhang et al. (2021c;b;a), representativeness is assumed to be related to the influence of nodes, and thus nodes with a larger influence score are first selected. Another line of work directly sets the accuracy of trained models as the objective (Gao et al., 2018; Hu et al., 2020a; Zhang et al., 2022), and then adopts reinforcement learning to do the optimization.

**LLMs for graphs.** Recent progress on applying LLMs for graphs (He et al., 2023a; Guo et al., 2023) aims to utilize the power of LLMs and further boost the performance of graph-related tasks. LLMs are either adopted as the predictor (Chen et al., 2023; Wang et al., 2023a; Ye et al., 2023), which directly generates the solutions or as the enhancer (He et al., 2023a), which takes the capability of LLMs to boost the performance of a smaller model with better efficiency. In this paper, we adopt LLMs as annotators, which combine the advantages of these two lines to train an efficient model with promising performance and good efficiency, without the requirement of any ground truth labels.

## 6 CONCLUSION

In this paper, we revisit the long-term ignorance of the data annotation process in existing node classification methods and propose the pipeline **label-free node classification on graphs with LLMs** to solve this problem. The key design of our pipelines involves LLMs to generate confidence-aware annotations, and using difficulty-aware selections and confidence-based post-filtering to further enhance the annotation quality. Comprehensive experiments validate the effectiveness of our pipeline.

## 7 ACKNOWLEDGEMENTS

This research is supported by the National Science Foundation (NSF) under grant numbers CNS 2246050, IIS1845081, IIS2212032, IIS2212144, IOS2107215, DUE 2234015, DRL 2025244 and IOS2035472, the Army Research Office (ARO) under grant number W911NF-21-1-0198, the Home Depot, Cisco Systems Inc, Amazon Faculty Award, Johnson&Johnson, JP Morgan Faculty Award and SNAP.

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

## A  DETAILED INTRODUCTIONS OF BASELINE MODELS

In this part, we present a more detailed introduction to related works, especially those baseline models adopted in this paper.

### A.1  GRAPH ACTIVE LEARNING METHODS

1. Density-based selection (Ma et al., 2022): We apply KMeans clustering in the feature space with the number of clusters set to the number of budgets. Then, nodes with closest distances to their clustering centers are selected as the training nodes. $f_{act}()$ is calculated as the reciprocal of the distance to the clustering centers.
2. GraphPart (Ma et al., 2022): A graph partition method is first applied to ensure the selection diversity. Then inside each region, nodes with high aggregated density are selected based on an approximate K-medoids algorithm. $f_{act}()$ is calculated as the reciprocal of the distance to the local clustering centers after partition.
3. FeatProp (Wu et al., 2019): Node features are first aggregated based on the adjacency matrix. Then, K-Medoids is applied to select the training nodes. $f_{act}()$ is calculated using K-Medoids.
4. Degree-based selection (Ma et al., 2022): Nodes with the highest degrees are selected as the training nodes. $f_{act}()$ is calculated as the degree od nodes.
5. Pagerank-based selection (Ma et al., 2022): Nodes with the highest PageRank centrality are selected as the training nodes. $f_{act}()$ is calculated as the PageRank score of nodes.
6. AGE (Cai et al., 2017): Nodes are selected based on a score function that considers both the density of features and PageRank centrality. We ignore the uncertainty measurement since we adopt a one-step selection setting. $f_{act}()$ is calculated as the weighted aggregation of the density of aggregated features together with the PageRank score.
7. RIM (Zhang et al., 2021b): Nodes with the highest reliable social influence are selected as the training nodes. Reliability is measured based on a pre-defined oracle accuracy and similarity between the aggregated node feature matrices. $f_{act}()$ is based on the density of aggregated features.

### A.2  ZERO-SHOT CLASSIFICATION BASELINE MODELS

1. SES (Li & Hooi, 2023): An unsupervised method to do classification. Both node features and class names are projected into the same semantic space, and the class with the smallest distance is selected as the label.
2. TAG-Z (Li & Hooi, 2023): Adopting prompts and graph topology to produce preliminary logits, which are readily applicable for zero-shot node classification.
3. BART-Large-MNLI (Lewis et al., 2019): A zero-shot text classification model fine-tuned on the MNLI dataset.

## B  MORE RELATED WORKS

**LLMs as Annotators.**  Curating human-annotated data is both labor-intensive and expensive, especially for intricate tasks or niche domains where data might be scarce. Due to the exceptional zero-shot inference capabilities of LLMs, recent literature has embraced their use for generating pseudo-annotated data (Bansal & Sharma, 2023; Ding et al., 2022; Gilardi et al., 2023; He et al., 2023b; Pangakis et al., 2023; Pei et al., 2023). Gilardi et al. (2023); Ding et al. (2022) evaluate LLMs' efficacy as annotators, showcasing superior annotation quality and cost-efficiency compared to human annotators, particularly in tasks like text classification. Furthermore, Pei et al. (2023); He et al. (2023b); Bansal & Sharma (2023) delve into enhancing the efficacy of LLM-generated annotations: Pei et al. (2023); He et al. (2023b) explore prompt strategies, while Bansal & Sharma (2023) investigates sample selection. However, while acknowledging their effectiveness, Pangakis et al. (2023) highlights certain shortcomings in LLM-produced annotations, underscoring the need for caution when leveraging LLMs in this capacity.

**Zero-shot Node Classification.**  Another issue related to our work is zero-shot node classification. The goal of zero-shot node classification is to train a model on existing training data and then generalize it to new categories. It is important to note that this is different from the concept of label-

free node classification that we propose, as in our problem there are no labels available from the start. Wang et al. (2021) transfers the knowledge from existing labels to unseen labels by semantic descriptions of labels. Yue et al. (2022) adopts a label consistency module to transfer the knowledge from the original label domain to the unseen label domain. Ju et al. (2023) adopts a two-level contrastive learning to learn the node embeddings and class assignments in an end-to-end manner, which can thus transfer the knowledge to unseen classes.

## C DATASETS

In this paper, we use the following popular datasets commonly adopted for node classifications: CORA, CITESEER, PUBMED, WIKICS, OGBN-ARXIV, and OGBN-PRODUCTS. Since LLMs can only understand the raw text attributes, for CORA, CITESEER, PUBMED, OGBN-ARXIV, and OGBN-PRODUCTS, we adopt the text attributed graph version from Chen et al. (2023). For WIKICS, we get the raw attributes from `https://github.com/pmernyei/wiki-cs-dataset`. Then, we give a detailed description of each dataset in Table 4.

Table 4: Dataset descriptions

| Dataset Name | #Nodes | #Edges | Task Description | Classes |
|---|---|---|---|---|
| CORA | 2708 | 5429 | Given the title and abstract, predict the category of this paper | Rule Learning, Neural Networks, Case Based, Genetic Algorithms, Theory, Reinforcement Learning, Probabilistic Methods |
| CITESEER | 3186 | 4277 | Given the title and abstract, predict the category of this paper | Agents, Machine Learning, Information Retrieval, Database, Human Computer Interaction, Artificial Intelligence |
| PUBMED | 19717 | 44335 | Given the title and abstract, predict the category of this paper | Diabetes Mellitus Experimental, Diabetes Mellitus Type 1, Diabetes Mellitus Type 2 |
| WIKICS | 11701 | 215863 | Given the contents of the Wikipedia article, predict the category of this article | Computational linguistics, Databases, Operating systems, Computer architecture, Computer security, Internet protocols, Computer file systems, Distributed computing architecture, Web technology, Programming language topics |
| OGBN-ARXIV | 169343 | 1166243 | Given the title and abstract, predict the category of this paper | 40 classes from `https://arxiv.org/archive/cs` |
| OGBN-PRODUCTS | 2449029 | 61859140 | Given the product description, predict the category of this product | 47 classes from Amazon, including Home & Kitchen, Health & Personal Care... |

## D PROMPTS

In this part, we show the prompts designed for annotations. We require LLMs to output a Python dictionary-like object so that it's easier to extract the results from the output texts. We find that LLMs sometimes generate information without following the users' instructions. As a result, we design the self-correction prompt to fix those invalid outputs (Table 5).

Table 5: Prompt used for self-correction

Previous prompt: **(Previous input)**
Your previous output doesn't follow the format, please correct it
old output: **(Previous output)**
Your previous answer **(Previous answer)** is not a valid class.
Your should only output categories from the following list:
**(Lists of label names)**
New output here:

We then demonstrate two concrete examples of our prompts. The first prompt is based on the zero-shot consistency strategy (see Table 6). The second prompt is based on the few-shot consistency strategy (see Table 7). It should be noted that the latter prompt is only used to be compared with zero-shot prompts. We don't use it in Section 4. For one-shot prompts, we leverage large language models to automatically generate both the top K predictions and confidence scores according to the philosophy of Zhang et al. (2023).

Table 6: Full prompt example for zero-shot annotation with consistency strategy on the PUBMED dataset

---

**Input:**  Question: **(Contents)** Paper: Title: Heritability of pancreatic beta-cell function among nondiabetic members of Caucasian familial type 2 diabetic kindreds. Abstract: Both defective insulin secretion and insulin resistance have been reported in relatives of type 2 diabetic subjects. We tested 120 members of 26 families with a type 2 diabetic sibling pair with a tolbutamide-modified, frequently sampled i.v....
Task:
There are following categories:
[diabetes mellitus experimental, diabetes mellitus type 1, diabetes mellitus type 2]
What's the category of this paper?
Provide your 3 best guesses and a confidence number that each is correct (0 to 100) for the following question from most probable to least. The sum of all confidence should be 100. For example, [ {"answer": <your_first_answer>, "confidence": <confidence_for_first_answer>}, ... ]
Output:

---

Table 7: Full prompt example for 1-shot annotation with TopK strategy on the CITESEER dataset

---

**Input:** I will first give you an example and you should complete task following the example.
Question: **(Contents for in-context learning samples)** Argument in Multi-Agent Systems Multi-agent systems ...**(Skipped for clarity)** Computational societies are developed for two primary reasons: Mode...
Task:
There are following categories:
[agents, machine learning, information retrieval, database, human computer interaction, artificial intelligence]
What's the category of this paper?
Provide your 3 best guesses and a confidence number that each is correct (0 to 100) for the following question from most probable to least. The sum of all confidence should be 100. For example, [ {"answer": <your_first_answer>, "confidence": <confidence_for_first_answer>}, ... ]
Output:
[{"answer": "agents", "confidence": 60}, {"answer": "artificial intelligence", "confidence": 30}, {"answer": "human computer interaction", "confidence": 10}]
Question: Decomposition in Data Mining: An Industrial Case Study Data **(Contents for data to be annotated)**
Task:
There are following categories:
[agents, machine learning, information retrieval, database, human computer interaction, artificial intelligence] What's the category of this paper?
Provide your 3 best guesses **(The same instruction, skipped for clarity)** ...
Output:

---

# E   COMPLETE RESULTS FOR THE PRELIMINARY STUDY ON EFFECTIVENESS OF CONFIDENCE-AWARE PROMPTS

The complete results for the preliminary study on confidence-aware prompts are shown in Table 8, Figure 6, and Figure 7.

Table 8: Accuracy of annotations generated by LLMs after applying various kinds of prompt strategies. We use yellow to denote the best and green to denote the second best result. The cost is determined by comparing the token consumption to that of zero-shot prompts.

| Prompt Strategy | CORA Acc (%) | Cost | CITESEER Acc (%) | Cost | PUBMED Acc (%) | Cost | OGBN-ARXIV Acc (%) | Cost | OGBN-PRODUCTS Acc (%) | Cost | WIKICS Acc (%) | Cost |
|---|---|---|---|---|---|---|---|---|---|---|---|---|
| Zero-shot | 68.33 ± 6.55 | 1 | 64.00 ± 7.79 | 1 | 88.67 ± 2.62 | 1 | 73.67 ± 4.19 | 1 | 75.33 ± 4.99 | 1 | 68.33 ± 1.89 | 1 |
| One-shot | 69.67 ± 7.72 | 2.2 | 65.67 ± 7.13 | 2.1 | 90.67 ± 1.25 | 2.0 | 69.67 ± 3.77 | 1.9 | 78.67 ± 4.50 | 1.8 | 72.00 ± 3.56 | 2.4 |
| TopK | 68.00 ± 6.38 | 1.1 | 66.00 ± 5.35 | 1.1 | 89.67 ± 1.25 | 1.1 | 72.67 ± 4.50 | 1 | 74.00 ± 5.10 | 1.2 | 72.00 ± 2.16 | 1.1 |
| Most Voting | 68.00 ± 7.35 | 1.1 | 65.33 ± 7.41 | 1.1 | 88.33 ± 2.49 | 1.4 | 73.33 ± 4.64 | 1.1 | 75.33 ± 4.99 | 1.1 | 69.00 ± 2.16 | 1.1 |
| Hybrid (Zero-shot) | 67.33 ± 6.80 | 1.5 | 66.33 ± 6.24 | 1.5 | 87.33 ± 0.94 | 1.7 | 72.33 ± 4.19 | 1.2 | 73.67 ± 5.25 | 1.4 | 71.00 ± 2.83 | 1.4 |
| Hybrid (One-shot) | 70.33 ± 6.24 | 2.9 | 67.67 ± 7.41 | 2.7 | 90.33 ± 0.47 | 2.2 | 70.00 ± 2.16 | 2.1 | 75.67 ± 6.13 | 2.3 | 73.67 ± 2.62 | 2.9 |

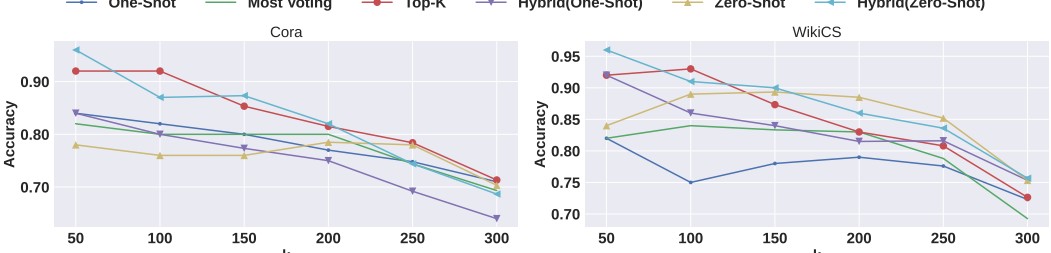

Figure 6: Relationship between the group accuracies sorted by confidence generated by LLMs. $k$ refers to the number of nodes selected in the groups.

## F PRELIMINARY OBSERVATIONS OF LLM'S ANNOTATIONS

### F.1 LABEL-LEVEL OBSERVATIONS

We begin by examining the label-level characteristics of annotations generated by LLMs. Label distribution is critical as it can influence model training. For example, imbalanced training labels may make models overfit to the majority classes (He & Garcia, 2009). Specifically, we juxtapose the distribution of LLM annotations against the ground truths. To facilitate this analysis, we employ noise transition matrices, a tool frequently used to study both real-world noisy labels (Wei et al., 2021) and synthetic ones (Zhu et al., 2021b). Considering a classification problem with $N$ classes, the noise transition matrix $T$ will be an $N \times N$ matrix where each entry $T_{ij}$ represents the probability that an instance from the true class $i$ is given the label $j$. If $i = j$, then $T_{ij}$ is the probability that a true class $i$ is correctly labeled as $i$. If $i \neq j$, then $T_{ij}$ is the probability that a true class $i$ is mislabeled as $j$.

To generate the noise transition matrices, we sample 1000 nodes randomly for each dataset. Figure 8 demonstrates the noise transition matrix for WIKICS, CORA, and CITESEER, three datasets with proper number of classes for visualization. We demonstrate the results of adopting the "zero-shot hybrid" prompting strategy in this figure. We illustrate the results for few-shot demonstration strategies and synthetic noisy labels in Appendix F.1. Since the number of samples in each class is imbalanced, we also show the label distribution for both ground truth and LLM's annotations in Figure 11.

An examination of Figure 8 reveals intriguing patterns. **FIRST**, the quality of annotations, defined as the proportion of annotations aligned with the ground truth, exhibits significant variation across different classes. For instance, in the WIKICS dataset, LLMs produce perfect annotations for samples where the ground truth class is $c_0$ (referring to "Computational linguistics"). In contrast, only 31% of annotations are accurate for samples belonging to $c_7$, referring to "distributed computing architecture". Similar phenomena can be observed in the CORA and CITESEER datasets. **SECOND**, for those classes with low annotation quality, incorrect annotations will not be distributed to other classes with a uniform probability. Instead, they are more likely to fall into one or two specific categories.

For example, for WIKICS, $c_7$ ("distributed computing architecture") tends to flip into $c_8$ ("web technology"). The reason may be that these two classes present similar meanings, and for some samples, categorizing them into any of these two classes is reasonable. Moreover, we find that such

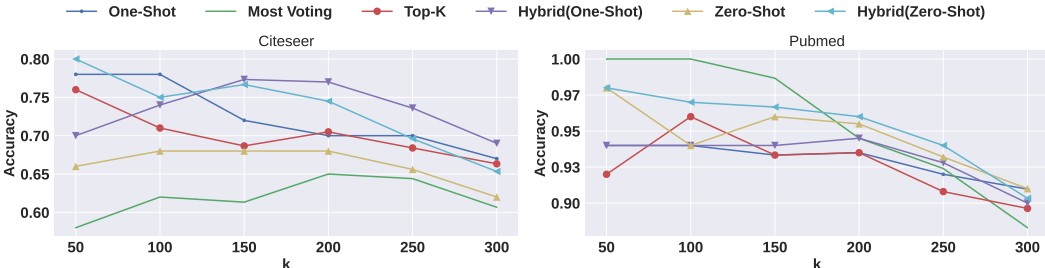

Figure 7: Relationship between the group accuracies sorted by confidence generated by LLMs. $k$ refers to the number of nodes selected in the groups.

kind of flipping can be asymmetric. Although $c_7$ in WIKICS tends to flip into $c_8$, samples from $c_8$ never flip into $c_7$. These properties make LLMs' annotations highly different from the synthetic noisy labels commonly adopted in previous literature (Song et al., 2022), which are much more uniform among different classes.

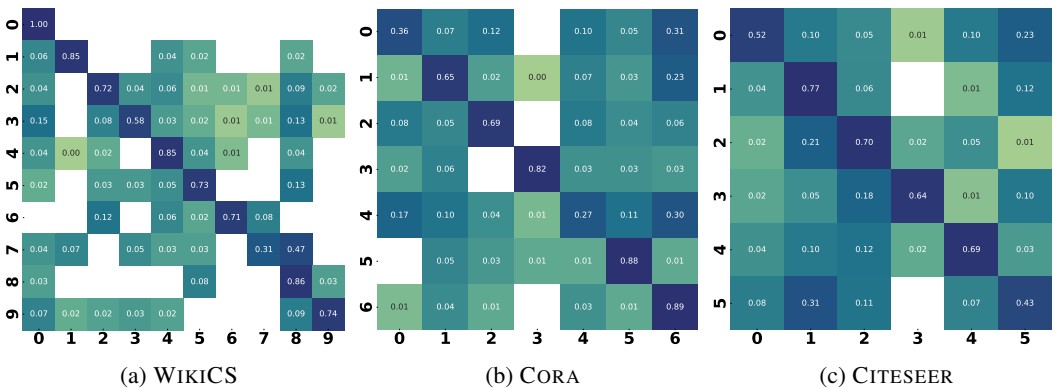

Figure 8: Noise transition matrix plot for WIKICS, CORA, and CITESEER with annotations generated by LLMs

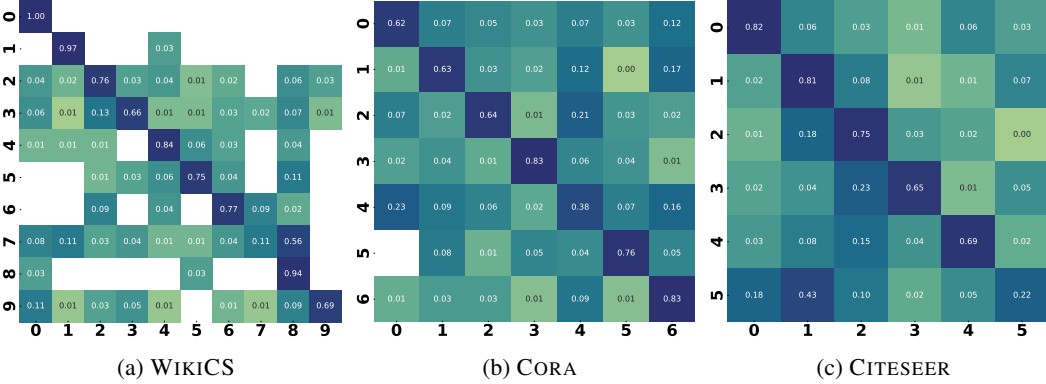

Figure 9: Noise transition matrix plot for WIKICS, CORA, and CITESEER with annotations generated by LLMs using prompts with few-shot demonstrations

Referring to Figure 11, we observe a noticeable divergence between the annotation distribution generated by LLMs and the original ground truth distribution across various datasets. Notably, in the WIKICS dataset, what is originally a minority class, $c_8$, emerges as one of the majority classes. A similar shift is evident with class $c_6$ in the CORA dataset. This trend can be largely attributed to the asymmetry inherent in the noise transition matrix.

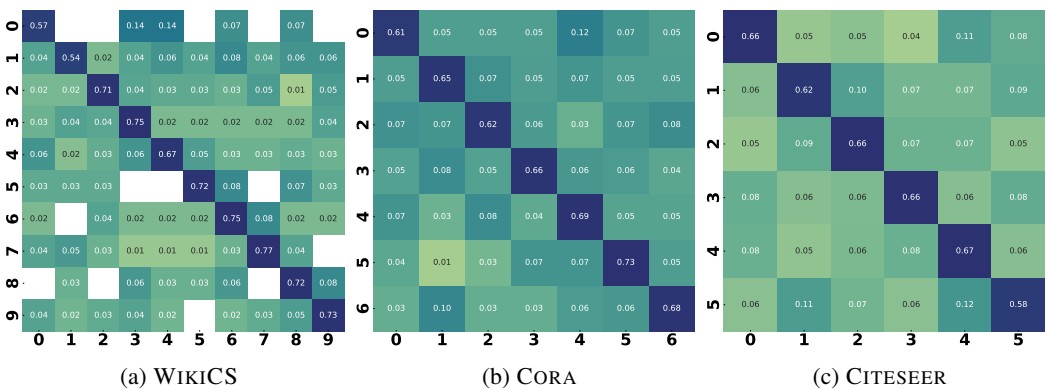

Figure 10: Noise transition matrix plot for WIKICS, CORA, and CITESEER with annotations generated by injecting random noise into the ground truth.

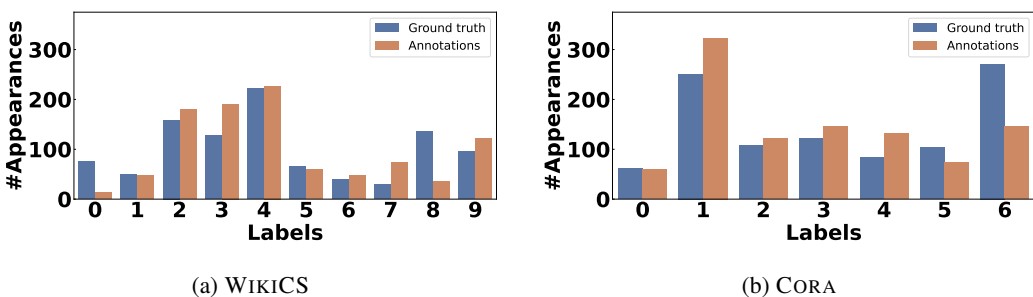

Figure 11: Label distributions for ground truth labels and LLM's annotations

## F.2 RELATIONSHIP BETWEEN ANNOTATION QUALITY AND C-DENSITY

As shown in Figure 12, the phenomenon on PUBMED is not as evident as on the other datasets. One possible reason is that the annotation quality of LLM on PUBMED is exceptionally high, making the differences between different groups less distinct. Another possible explanation, as pointed out in (Chen et al., 2023), is that LLM might utilize some shortcuts present in the text attributes during its annotation process, and thus the correlation between features and annotation qualities is weakened.

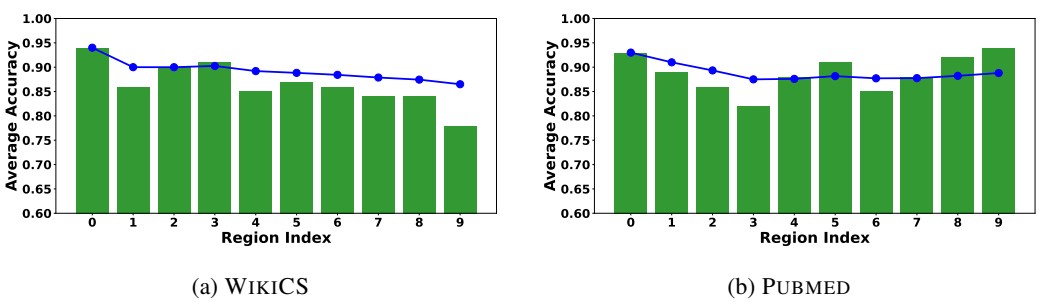

Figure 12: Relationship between the group accuracies and distances to the clustering centers. The bar shows the average accuracy inside the selected group. The line shows the accumulated average accuracy.

## G    EFFECTIVENESS OF CONFIDENCE GENERATED BY LLMS

In this section, we demonstrate the effectiveness of confidence generated by LLMs with a case study on the Cora dataset. We plot the confidence calibration plot with three prompt strategies: zero-shot, TopK, and hybrid. From the results, we can see that hybrid prompt can generate accurate and diverse confidence scores, which is more effective.

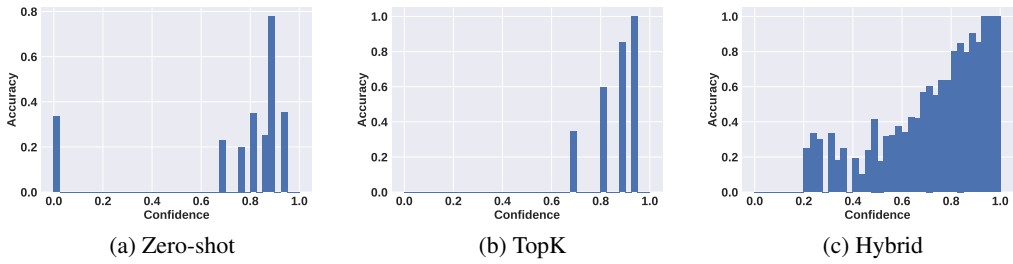

(a) Zero-shot          (b) TopK          (c) Hybrid

Figure 13: Comparison of three prompt strategies

## H    HYPERPARAMTERS

We now demonstrate the hyper-parameters adopted in this paper, which is inspired by  Hu et al. (2020b):

1. For small-scale datasets including CORA, CITESEER, PUBMED, and WIKICS, we set: learning rate to $0.01$, weight decay to $5e^{-4}$, hidden dimension to $64$, dropout to $0.5$.
2. For large-scale datasets including OGBN-ARXIV and OGBN-PRODUCTS, we set: learning rate to $0.01$, weight decay to $5e^{-4}$, hidden dimension to $256$, dropout to $0.5$.

## I    COMPARING LLMS ANNOTATIONS TO SYNTHETIC NOISY LABELS

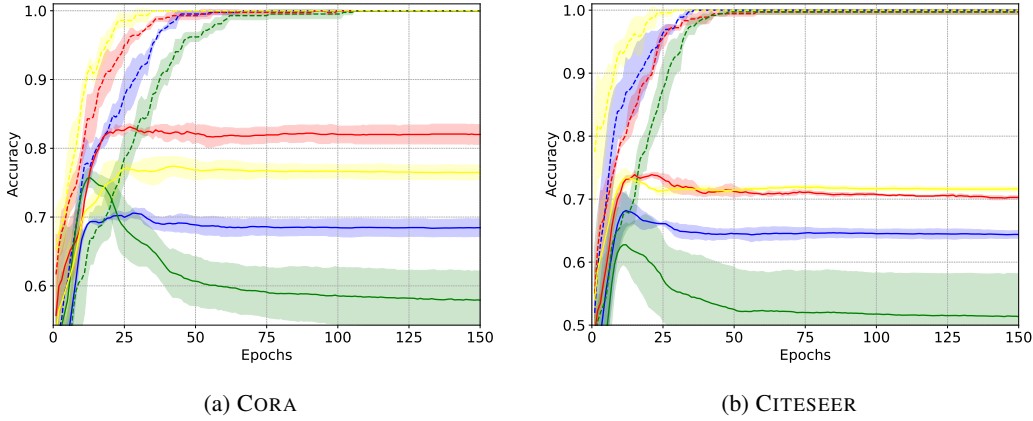

(a) CORA          (b) CITESEER

Figure 14: The red curve represents the performance of models trained with ground truth labels. The yellow curve represents the performance of models trained with LLMs' annotations with all wrong labels fixed. The blue curve represents the models trained by LLMs' annotations. The green curve represents the models trained by synthetic noisy labels with the same annotation quality as LLMs' annotations. The solid line represents the performance on the test set, while the dashed line represents the performance on the training set.

## J    EXTRA RESULTS FOR THE COMPARATIVE STUDY IN TABLE 2

In Table 10, we further demonstrate the results for combing weighted loss, difficulty-aware selection, and post-filtering to degree-based selection and pagerank-based selection. In Table 9, we explore the effectiveness of applying PS and DA together. We find that applying them simultaneously usually won't get good performance, which means it's probably not the proper way to integrate LLMs' confidence into selection. As a comparison, we find that using weighted cross-entropy loss can enhance the performance most of the time. Even though it can not surpass the baselines, the gap is very small, which shows its effectiveness.

Table 9: Ablation study for the effectiveness of different combinations.

|  | Cora | CiteSeer |
|---|---|---|
| AGE | $69.15 \pm 0.38$ | $54.25 \pm 0.31$ |
| DA-AGE | $74.38 \pm 0.24$ | $59.92 \pm 0.42$ |
| DA-AGE-W | $74.96 \pm 0.22$ | $58.41 \pm 0.45$ |
| PS-DA-AGE | $71.53 \pm 0.19$ | $56.38 \pm 0.14$ |
| RIM | $69.86 \pm 0.38$ | $63.44 \pm 0.42$ |
| DA-RIM | $73.99 \pm 0.44$ | $60.33 \pm 0.40$ |
| DA-RIM-W | $74.73 \pm 0.41$ | $60.80 \pm 0.57$ |
| PS-DA-RIM | $72.34 \pm 0.19$ | $60.33 \pm 0.40$ |

Table 10: Extra results for degree-based selection and pagerank-based selection

|  | CORA | CITESEER | PUBMED | WIKICS | OGBN-ARXIV | OGBN-PRODUCTS |
|---|---|---|---|---|---|---|
| Degree | $68.67 \pm 0.30$ | $60.23 \pm 0.54$ | $67.77 \pm 0.07$ | $65.38 \pm 0.35$ | $54.98 \pm 0.37$ | $71.22 \pm 0.21$ |
| Degree-W | $69.86 \pm 0.35$ | $60.47 \pm 0.49$ | $68.24 \pm 0.09$ | $65.61 \pm 0.31$ | $55.69 \pm 0.24$ | $71.96 \pm 0.23$ |
| DA-Degree | $72.86 \pm 0.27$ | $60.23 \pm 0.54$ | $74.51 \pm 0.04$ | $63.40 \pm 0.51$ | $55.32 \pm 0.33$ | $44.41 \pm 0.53$ |
| PS-Degree-W | $70.92 \pm 0.28$ | $62.36 \pm 0.69$ | $74.83 \pm 0.05$ | $67.21 \pm 0.29$ | $55.89 \pm 0.39$ | $71.57 \pm 0.25$ |
| DA-Degree-W | $73.01 \pm 0.24$ | $61.29 \pm 0.47$ | $74.11 \pm 0.04$ | $63.14 \pm 0.55$ | $55.35 \pm 0.32$ | $47.90 \pm 0.45$ |
| Pagerank | $70.31 \pm 0.42$ | $61.21 \pm 0.11$ | $68.58 \pm 0.14$ | $67.13 \pm 0.46$ | $59.52 \pm 0.03$ | $69.20 \pm 0.32$ |
| Pagerank-W | $71.50 \pm 0.44$ | $61.97 \pm 0.19$ | $68.86 \pm 0.19$ | $69.61 \pm 0.34$ | $59.60 \pm 0.04$ | $69.75 \pm 0.29$ |
| DA-Pagerank | $74.34 \pm 0.41$ | $60.44 \pm 0.40$ | $72.84 \pm 0.15$ | $67.15 \pm 0.44$ | $58.82 \pm 0.52$ | $54.77 \pm 0.36$ |
| PS-Pagerank-W | $74.81 \pm 0.37$ | $63.27 \pm 0.34$ | $68.23 \pm 0.17$ | $69.86 \pm 0.29$ | $58.84 \pm 0.14$ | $69.69 \pm 0.45$ |
| DA-Pagerank-W | $75.62 \pm 0.39$ | $61.25 \pm 0.45$ | $73.60 \pm 0.22$ | $68.19 \pm 0.32$ | $59.40 \pm 0.26$ | $55.57 \pm 0.24$ |

## K    THEORETICAL MOTIVATION

In this section, we further analyze the theoretical motivation for difficulty-aware selection. In a nutshell, our objective is to show **why C-Density is a useful metric to select nodes with high annotation quality.**

Given the parameter distribution of LLMs $\mathcal{Q}$, the parameter distribution of encoder $\mathcal{P}$ (SBERT), we assume ground truth $Y_L \in \mathcal{R}^{N \times M}$, pseudo label $Y \in \mathcal{R}^{N \times M}$, and node features encoded by the encoder $X \in \mathcal{R}^{N \times d}$. Here, $M$ denotes the number of classes and $d$ denotes the hidden dimension. Our objective is to maximize the accuracy of annotations, which is thus to minimize the discrepancies between $Y$ and $Y_L$.

$$\min_{(n_1,...,n_k)} \mathbb{E}_{\theta \sim Q} f(Y) = \ell(Y, Y_L) = \sum_{i=1}^{k} \ell(y^{n_i}, y_L^{n_i})$$

where $(n_1, ..., n_k)$ is the index of the $k$ selected nodes, and $f(Y)$ is defined as follows: $x^{n_i}$ represents the feature of node $n_i$, while $x_L^{n_i}$ represents the unknown latent embedding of node $n_i$.

$$f(Y) = \ell(Y, Y_L) = \sum_{i=1}^{k} \ell(y^{n_i}, y_L^{n_i}) = \ln(1 + \sum_{i=1}^{k} \|x^{n_i} - x_L^{n_i}\|^2) = \ln(1 + \|X - X_L\|^2)$$

Since we only have access to the node features $X$ generated by encoder $\theta$, we need to make a connection between $\mathcal{Q}$ and $\mathcal{P}$.

**Lemma 1** *For any annotation $y$ generated by LLMs, $\mathbb{E}_{\theta \sim \mathcal{Q}} f(y) \leq \log \mathbb{E}_{\theta' \sim \mathcal{P}} \exp(f(y)) + KL(\mathcal{Q}\|\mathcal{P})$.*

**Proof.**

$$\mathbb{E}_{\theta' \sim \mathcal{P}} f(y) = \int f(y) p(y) dy = \int f(y) \frac{p(y)}{q(y)} q(y) dy = \mathbb{E}_{\theta \sim \mathcal{Q}} f(y) \frac{p(y)}{q(y)}$$

Since $KL(\mathcal{Q}\|\mathcal{P}) = \mathbb{E}_{\theta \sim Q} \log \frac{q(y)}{p(y)}$,

$$\log \mathbb{E}_{\theta' \sim \mathcal{P}} f(y) = \log \mathbb{E}_{\theta \sim \mathcal{Q}} f(y) \frac{p(y)}{q(y)} \geq \mathbb{E}_{\theta \sim \mathcal{Q}} \log f(y) \frac{p(y)}{q(y)} = \mathbb{E}_{\theta \sim \mathcal{Q}} \log f(y) - KL(\mathcal{Q}\|\mathcal{P})$$

Then, we transform the objective into

$$\min_{(n_1, \ldots, n_k)} \mathbb{E}_{\theta' \sim \mathcal{P}} \exp(f(Y))$$

Assuming the label distribution $\mathcal{H}$, we further have

$$\min_{(n_1, \ldots, n_k)} \mathbb{E}_{Y \sim \mathcal{H}} \mathbb{E}_{\theta' \sim \mathcal{P}} \exp(f(Y)) = \min_{(n_1, \ldots, n_k)} \mathbb{E}_{Y \sim \mathcal{H}} \mathbb{E}_{\theta \sim \mathcal{P}} \|X - X_L\|^2$$

Assuming $X$ and $X_L$ follows gaussian distribution, where $X \sim N(\mu_i, \sigma_i); X_L \sim N(\mu_j, \sigma_j)$, then

$$\begin{aligned}
\mathrm{E}\left(\|X - X_L\|^2\right) &= \mathrm{E}\left(\|(X - \mu_i) - (X_L - \mu_j) + (\mu_i - \mu_j)\|^2\right) \\
&= \mathrm{E}\left(\|X - \mu_i\|^2\right) + \mathrm{E}\left(\|X_L - \mu_j\|^2\right) + \|\mu_i - \mu_j\|^2 \\
&= n\sigma_i^2 + n\sigma_j^2 + \|\mu_i - \mu_j\|^2
\end{aligned}$$

Since $(\mu_j, \sigma_j)$ is unknown, $\mu_i$ is fixed, thus we want to minimize $\sigma_i$. Given an arbitrary node, $\sigma_i$ can be viewed as the distance to the clustering centers. The smaller $\sigma_i$ is, the smaller the corresponding $\mathrm{E}\left(\|X - X_L\|^2\right)$ also becomes, indicating that the minimum value of $\mathbb{E}_{\theta \sim \mathcal{Q}} f(y)$ is attained when $\sigma_i$ is at its smallest. This demonstrates why nodes closer to the clustering centers are "preferred" by LLMs and thus achieve better annotation quality.

## L    DESIGN PHILOSOPHY BEHIND LLMGNN

Regarding LLMGNN, we have the following two key designs:

1. In the annotation process, we do not consider structural information, hence we have designed a structure-free prompt. 2. In the choice of a LLM, we do not use the more advanced GPT-4, but instead, we choose the more cost-effective GPT-3.5-turbo. In practice, we find that these two designs are currently the most appropriate because: 1. We discover that using a structure-aware prompt does not effectively improve annotation quality without introducing ground truth labels, due to the limited structural understanding capabilities of LLMs. 2. We find that compared to GPT-3.5-turbo, the improvement brought by GPT-4 is very limited. After exploration, we realize that this is related to the ambiguity in the ground truth labels of the node classification task. Currently, GNN is the best tool capable of utilizing structural information to handle ambiguity.

To begin with, assigning correct labels for node classification is hard for LLMs. We show the **annotation quality** (the ratio of annotations matching the ground truth labels) of GPT3.5 and GPT4 in Table 11 by **randomly** selecting the annotated samples (140 nodes for Cora, and 120 nodes for CiteSeer, and we control the seed to make sure we select the same set of nodes). We can see that GPT4 doesn't give much better annotations (in terms of accuracy) than GPT3.5. The overall performance (less than 70%) indicates that node classification is not an easy task for LLMs.

If we further check the annotation results given by these two different models, we find that one common bottleneck preventing these models from getting better results is the **annotation bias** (also can be viewed as "**label ambiguity**") of the labels. We showcase one example in Table 12.

Table 11: Comparison of GPT-3.5 and GPT-4 on the annotation task

|  | CORA | CITESEER |
|---|---|---|
| **GPT3.5** | 67.62 ± 2.05 | 66.39 ± 8.62 |
| **GPT4** | 68.81 ± 1.87 | 65.56 ± 9.29 |

Table 12: An example from the CITESEER dataset

**Input:** Attribute of one node from Citeseer: 'Discovering Web Access Patterns and Trends by Applying OLAP and Data Mining Technology on Web Logs As a confluence of data mining and WWW technologies, it is now possible to perform data mining on web log records collected from the Internet web page access history...

The ground truth label for this node is "Database", while the prediction of both two LLMs is "Information Retrieval". From human beings' understanding, both "Database" and "Information retrieval" are somewhat reasonable categories of this node. However, because of the single label setting of node classification, only one of them is correct. This somehow demonstrates that to get high accuracy on node classification, the models need to capture such kinds of **dataset-specific bias** in the annotation (the information in LLMs is more like "commonsense knowledge", so there can be a mismatch). Structural information may help us find such kind of bias. For example, if the category of neighboring nodes is more easier to predict, and most of them should be related to "Database", then this node is more likely to come from "Database". This phenomenon has also been observed in Chen et al. (2023).

So, why LLMs with structure-aware prompts can not improve their performance by introducing that structural information (shown in Table 13)? It's mainly because of LLMs' poor understanding capability of structural information. Huang et al. (2023) and Wang et al. (2023a) try various kinds of structure-aware prompts, and find that LLMs present limited structure reasoning abilities with current prompt designs. Chen et al. (2023) and Huang et al. (2023) further show that incorporating **neighboring ground truth labels** can improve the performance. However, ground truth labels are not available in the label-free settings. As a comparison, we find that merely summarizing the neighboring contents may not effectively improve the performance. Under the current structure-aware prompt design, LLMs can only utilize the structural information in a very limited manner (like using neighboring labels). As a comparison, message-passing GNNs can capture complicated structural information effectively and efficiently. For example, we try replacing GNN with MLP in LLMGNN in the Table 14, and we observe a huge performance gap.

Table 13: Comparison of using structure-aware and structure-free prompts

|  | CORA | CORA | CITESEER | CITESEER |
|---|---|---|---|---|
|  | No Struct | Struct | No Struct | Struct |
| FeatProp | 72.82 ± 0.08 | 69.08 ± 0.39 | 66.61 ± 0.55 | 65.92 ± 0.43 |
| FeatProp + PS | 75.54 ± 0.34 | 67.55 ± 0.70 | 69.06 ± 0.32 | 67.80 ± 0.45 |

Based on the above reasons, we only use textual information in the labeling process, while in the training process of GNN, we utilize structural information. At the current stage, we believe this to be a more effective paradigm. This highlights the motivation for us to design LLMGNN which can enjoy the advantages of both LLMs and GNNs while mitigating their limitations.

Designing an effective structure-aware prompt is a valuable future direction, and methods like Agent-based prompting (Wang et al., 2023b) (multi-round prompt) may help to further improve the performance. The main focus of our paper is to propose a flexible framework that supports various kinds of prompt designs, and new prompt designs can also enhance the effectiveness of our framework.

Table 14: Comparisons of LLMs-as-Predictors, LLMGNN, and LLMMLP, which demonstrates the superiority of GNN

|       | LLMs-as-Predictors | LLMGNN | LLMMLP |
|-------|--------------------|--------|--------|
| CORA  | 68.33              | 76.23  | 67.06  |

The importance of the prompt is reflected in the fact that if we can improve the quality of annotations, we can further enhance the effectiveness of LLMGNN. For example, if we correct a portion of incorrect annotations to the right ones, we may improve the performance of LLMGNN. The "original quality" means the original annotation given by LLMs. "+k%" means that we randomly turn "k%" wrong annotations into correct ones. The results are shown in Table 15.

Table 15: The performance-changing trend of Random selection and PS-FeatProp-W selection when fix a portion of the wrong annotations

|               | original quality | +7%   | +12%  | +17%  |
|---------------|------------------|-------|-------|-------|
| Random        | 70.48            | 72.66 | 75.92 | 78.58 |
| PS-FeatProp-W | 76.23            | 77.95 | 80.25 | 81.96 |

Moreover, we want to show that the annotation quality will influence the scaling behavior of LLMGNN when we increase the budget. From Table 16 (the first row means the budget equal to $k \times$ number of classes ), we can see that (1) When the budget is low, the difference between LLMGNN trained by high-quality and low-quality annotations is smaller ($< 20$); when the budget is high, this difference becomes larger($> 20$); (2) the performance of LLMGNN is closely related to the annotation quality (the performance of LLMGNN on PubMed is better than one on Cora). With better annotation quality, LLMGNN may potentially achieve a higher performance upper bound when we gradually increase the budgets.

Table 16: The scaling behavior of LLMGNN on CORA and PUBMED when we increase the budgets

|        | 5     | 10    | 15    | 20    | 25    | 40    | 80    | 160   |
|--------|-------|-------|-------|-------|-------|-------|-------|-------|
| CORA   | 57.35 | 66.45 | 68.42 | 70.17 | 69.64 | 70.68 | 72.07 | 72.73 |
| PUBMED | 62.49 | 64.56 | 72.44 | 73.16 | 75.92 | 77.8  | 82.38 | 82.5  |

