# OpenReview forum: "Label-free Node Classification on Graphs with Large Language Models (LLMs)"
_ICLR.cc/2024/Conference — ICLR 2024 poster_

### Official Review · Reviewer_LGrX · 2023-10-29

**Soundness:** 4 excellent
**Presentation:** 3 good
**Contribution:** 4 excellent
**Rating:** 8
**Confidence:** 5

**Summary:**

The paper introduces a novel approach called LLM-GNN for label-free node classification on graphs, which combines the strengths of Graph Neural Networks (GNNs) and Large Language Models (LLMs) while mitigating their limitations. It addresses the challenge of obtaining high-quality labels for graph-structured data by leveraging LLMs' zero-shot learning capabilities. LLM-GNN actively selects nodes for annotation by LLMs, generates confidence-aware annotations, and refines annotation quality through post-filtering. The approach achieves impressive results on a massive-scale dataset, OGBN-PRODUCTS, without the need for costly human annotations.

**Strengths:**

1. LLM-GNN presents an innovative approach to node classification on graphs by harnessing the complementary strengths of GNNs and LLMs. It acknowledges the challenges of obtaining high-quality labels and proposes a label-free solution, which is a significant contribution to the field of machine learning.
2. The paper demonstrates the cost-effectiveness of LLM-GNN by achieving high accuracy on a large dataset with annotation costs under 1 dollar. This cost-efficient approach is particularly relevant for real-world applications with resource constraints.
3. LLM-GNN offers a comprehensive methodology that not only utilizes LLMs for annotations but also considers active node selection, confidence-aware annotations, and post-filtering. This approach ensures the quality, representativeness, and diversity of annotations, addressing key challenges in label-free node classification.

**Weaknesses:**

1. Since LLMs generate annotations without access to ground truth labels, there is a risk of noisy annotations.  It would be better to investigate the robustness of LLM-GNN to noisy annotations and potential strategies for mitigating their effects.
2.  LLM-GNN's performance is demonstrated on a specific dataset (OGBN-PRODUCTS), and while it achieves impressive results, its generalizability to other datasets or domains is not thoroughly explored in the paper. The effectiveness of the approach in different scenarios and with various types of graphs should be investigated to assess its broader applicability.
3. There could be better with a detired comparison on the economic perspective.

**Questions:**

see the weakness

---

> ### Author Response · Authors · 2023-11-18
> **Response to Reviewer LGrX (1/3)**
>
> Q1: Since LLMs generate annotations without access to ground truth labels, there is a risk of noisy annotations. It would be better to investigate the robustness of LLM-GNN to noisy annotations and potential strategies for mitigating their effects.
>
> **Response** Thanks for your great comments. We'll first summarize our findings and show details in the next paragraph. (1) We find that normal cross-entropy loss has a smaller overfitting on LLMs' annotations compared to random noisy labels, which is one advantage of using LLMs as the annotators; (2) We find that (graph) noisy label learning methods do not show clear improvements compared to normal cross-entropy loss in our setting. **Simple normal and weighted cross-entropy loss works better**.
>
> (1) To show the characteristics of LLMs' annotations, we train GNN models with ground truth labels, LLMs' annotations, and synthetic noisy labels with the same annotation quality (for example, suppose LLMs correctly label 80% of the points, we will randomly replace the labels of 20% of the points with incorrect labels based on the ground truth labels. ) We put the training curves in Appendix J and show the maximum test accuracies and final test accuracies in the following table. We observe that, unlike synthetic noisy labels, the gap between maximum accuracy and final accuracy for LLMs is very small. On one hand, it shows one good point of LLMs' annotations that we don't use a validation set to find the stopping point. On the other, it implies that it may be difficult to design a loss function to improve the performance.
>
> |                        	|      Cora      	|       Cora       	|    CiteSeer    	|     CiteSeer     	|
> |----------------------|--------------|----------------|--------------|----------------|
> |                        	| Final accuracy 	| Maximum accuracy 	| Final accuracy 	| Maximum accuracy 	|
> |   Ground truth labels  	|      0.82      	|       0.825      	|       0.7      	|       0.73       	|
> |   LLMs's annotations   	|      0.705     	|       0.71       	|      0.65      	|       0.68       	|
> | Synthetic noisy labels 	|      0.58      	|       0.76       	|      0.51      	|       0.62       	|
>
> (2) We further compare the differences among different loss functions. We compare the following four baselines: random selection and normal cross-entropy loss; random selection and NCERCE loss (design for noisy label learning) [1]; RIM with normal cross-entropy loss; random selection and weighted cross entropy loss; original RIM with weighted loss. It should be noted that RIM is an active learning method designed to process synthetic noisy labels. From the table, we find that loss designed for noisy label learning doesn't present an advantage over normal and weighted cross-entropy loss. One possible reason for this behavior is that the noisy pattern of LLMs' annotations is much more complex than synthetic noisy labels (see Appendix G.1), while noisy label learning methods are usually evaluated on synthetic noisy labels. Another reason is that the number of training samples is limited so it's hard to capture the pattern of noisy annotations. Compared to noisy label learning loss, we find that a simple weighted cross entropy loss ("WE" in the table) works well in most cases.
>
> |                	|  Cora 	| CiteSeer 	|
> |--------------|-----|--------|
> |    Random+CE   	| 70.48 	|   65.11  	|
> | Random+NCE+RCE 	| 61.18 	|   67.75  	|
> |     RIM+CE     	| 67.39 	|   64.37  	|
> |    Random+WE	    | 71.85	    |   66.72   |
> |       RIM      	| 68.28 	|   63.06  	|
>
>
> [1] Ma X, Huang H, Wang Y, et al. Normalized loss functions for deep learning with noisy labels[C]//International conference on machine learning. PMLR, 2020: 6543-6553.

---

> ### Author Response · Authors · 2023-11-18
> **Response to Reviewer LGrX (2/3)**
>
> Q2: LLM-GNN's performance is demonstrated on a specific dataset (OGBN-PRODUCTS), and while it achieves impressive results, its generalizability to other datasets or domains is not thoroughly explored in the paper. The effectiveness of the approach in different scenarios and with various types of graphs should be investigated to assess its broader applicability.
>
> **Response** Thanks for your great comments. We are not quite sure whether we exactly understand some details of your remarks. We will first retell the key points and answer them accordingly.
>
> We think you make the following key points: (1) Only the results of OGBN-Products are demonstrated. (2) Whether the proposed pipeline can be extended to more tasks like graph classification and more types of graphs beyond text-attributed graphs needs to be explored.
>
> For the first point, we admit we focus on the performance of OGBN-Products in the abstract and introduction part, which may lead to confusion. **We also test five other datasets in the experiment part.** For the second question, we think our pipelines have the potential to be applied to **more types of graphs and more types of tasks**.
>
>
> (1) Other than the performance on OGBN-PRODUCTS, we also show the performance of our methods on other datasets like Cora, CiteSeer, Pubmed, WikiCS, and Arxiv in the paper, and we will show them in the following table. Specifically, Cora, CiteSeer, and Arxiv are about papers from the computer science domain. Pubmed is about the papers from the medical domain. WikiCS is about the Wikipedia page. We have included multiple datasets from different domains to show the general applicability of our methods. Moreover, our pipeline shows consistent effectiveness across these datasets. In the following table, we demonstrate that our pipelines can get good performance with a much lower cost (both money and time) and scale to large graphs.
>
> | Performance        	| Cora  	| CiteSeer 	| PubMed 	| WikiCS 	| Arxiv 	| Products 	|
> |--------------------|------|----------|--------|--------|-------|----------|
> | LLMGNN             	| 75.54 	| 69.06    	| 81.95  	| 66.09  	| 66.14 	| 74.91    	|
> | LLMs-as-Predictors 	| 67.33 	| 66.33    	| 87.33  	| 71     	| 73.67 	| 75.33    	|
>
> | Costs              	| Cora  	| CiteSeer 	| PubMed 	| WikiCS 	| Arxiv 	| Products 	|
> |--------------------|-------|----------|--------|--------|-------|----------|
> | LLMGNN             	| 0.11  	| 0.1      	| 0.25   	| 0.16   	| 0.63  	| 0.74     	|
> | LLMs-as-Predictors 	| 1.26  	| 1.5      	| 9.2    	| 5.46   	| 79    	| 1952     	|
>
> (2) We admit that our paper focuses on the node classification tasks on the text-attributed graphs. However, we think our pipeline has the potential to be extended to more types of tasks and graphs.
>
> For different tasks like graph classification and knowledge graph QA, [1] does some exploration, and LLMs also show promising zero-shot capabilities. As a result, we may extend our pipelines to these tasks with some small modifications to the architectures.
>
> For more types of graphs, in a recent paper [2], the authors introduced a prompt-based method that aligns all types of node features through natural language, allowing large language models to naturally process problems in the textual domain with notable success. For discrete features like bag of words, [3] also propose a potential way to let large language models handle them by converting the original features into clustering centers. These methods may be integrated with our work and extend our methods to more types of graphs. We think you make two valuable points for future directions.
>
> [1] Guo J, Du L, Liu H. GPT4Graph: Can Large Language Models Understand Graph Structured Data? An Empirical Evaluation and Benchmarking[J]. arXiv preprint arXiv:2305.15066, 2023.
>
> [2] Liu H, Feng J, Kong L, et al. One for All: Towards Training One Graph Model for All Classification Tasks[J]. arXiv preprint arXiv:2310.00149, 2023.
>
> [3] Zhao J, Zhuo L, Shen Y, et al. Graphtext: Graph reasoning in text space[J]. arXiv preprint arXiv:2310.01089, 2023.

---

> ### Author Response · Authors · 2023-11-18
> **Response to Reviewer LGrX (3/3)**
>
> Q3: There could be better with a detailed comparison from the economic perspective.
>
> **Response** Thanks for your great comments, which inspire us to further compare both the economy and performance of our pipelines. Compared to other baselines, our methods can **achieve promising performance with very low costs**.
>
>
> We use the following data to validate our claims. The first table demonstrates the performance of each pipeline. For LLMGNN and GNN, we deliberately let them produce similar performances to conduct a fair cost comparison. For Pubmed*, we demonstrate the performance of LLMGNN with a larger budget since the performance of LLMs-as-Predictors is superior on this dataset. We can improve the performance of LLMGNN by slightly increasing the costs.
> |     (Accuracy)     	|  Cora 	| CiteSeer 	| Pubmed 	| Pubmed* 	| WikiCS 	| Arxiv 	| Products 	|
> |------------------|-----|--------|------|-------|------|-----|--------|
> |       LLMGNN       	| 75.54 	|   69.06  	|  74.98 	|  81.95  	|  66.09 	| 66.14 	|   74.9   	|
> | LLMs-as-Predictors 	| 68.33 	|   66.33  	|  87.33 	|  87.33  	|   71   	| 73.33 	|   75.33  	|
> |         GNN        	| 75.54 	|   69.06  	|  74.98 	|  81.95  	|  66.09 	| 66.14 	|   74.9   	|
>
> The second table compares the number of training samples needed to achieve the performance in the first table. Here, PL stands for pseudo labels generated by LLMs. GT stands for ground truth labels.
> | (Number of   annotations) 	| Cora 	| CiteSeer 	| Pubmed 	| Pubmed* 	| WikiCS 	|  Arxiv 	| Products 	|
> |-------------------------|----|--------|------|-------|------|------|--------|
> |         LLMGNN (PL)        	|  140 	|    120   	|   60   	|   300   	|   200  	|   800  	|    940   	|
> |   LLMs-as-Predictors (PL)  	| 2708 	|   3186   	|  19717 	|  19717  	|  11701 	| 169343 	|  2449029 	|
> |          GNN (GT)         	|  50  	|    50    	|   42   	|   210   	|   40   	|   560  	|    400   	|
>
>
> From these two tables, we can see that **LLMGNN significantly reduces the costs compared to LLMs-as-Predictors**. Comparing GNN with LLMGNN, the costs of ground truth labels are hard to estimate. From a recent study [1], it shows that the annotations generated by ChatGPT are more than 20 times cheaper than human annotators. If we use this ratio, we can see that although LLMGNN requires more pseudo labels, the overall costs are still much cheaper than traditional GNN-based pipelines.
>
> [1] Gilardi F, Alizadeh M, Kubli M. Chatgpt outperforms crowd-workers for text-annotation tasks[J]. arXiv preprint arXiv:2303.15056, 2023.

---

> ### Comment · Reviewer_LGrX · 2023-11-20
> **Thanks for your response**
>
> Thanks for your response which solved all of my concerns. I have raised my score accordingly.

---

> > ### Author Response · Authors · 2023-11-20
> > **Thanks for your response**
> >
> > Thanks for your response and support. We are glad to know that our rebuttal has addressed your concerns. Please let us know in case there remain outstanding concerns, and if so, we will be happy to respond.

---

### Official Review · Reviewer_HZKD · 2023-10-30

**Soundness:** 3 good
**Presentation:** 3 good
**Contribution:** 2 fair
**Rating:** 6
**Confidence:** 4

**Summary:**

In this paper, the authors study label-free node classification by combining LLMs with GNNs. Specifically, the proposed method first leverages LLMs to annotate a small portion of nodes, then uses GNNs with the pseudo-labels to classify the remaining large portion of nodes. The main challenges lie in how to actively select nodes and leverage LLMs to obtain reliable labels for those nodes. Three modules, including difficulty-aware active node selection, confidence-aware annotations, and post-filtering are proposed to tackle the challenges. Experimental results on text-attributed graphs demonstrate the effectiveness of the proposed method, especially compared with heuristically choosing annotated nodes.

**Strengths:**

1. The proposed method is among the first trials of combining LLMs with GNNs to solve a novel problem, i.e., label-free node classification.
2. The proposed method is clearly described and the paper is easy to follow in general.
3. The authors compare with various heuristic baselines and conduct analyses to demonstrate the efficacy of the proposed method.

**Weaknesses:**

1.	Though I acknowledge that the proposed method is a valid solution, the technical contribution of the paper is somewhat limited, especially considering that the three major components are largely based on heuristic observations, and the rest are based on existing LLMs and GNNs. It would make the paper stronger if some theoretical analyses could be provided for the proposed components.
2.	The authors should more explicitly mention that their proposed method only works for text-attributed graphs rather than any general graph, e.g., in the abstract and introduction. Otherwise, the paper may have overclaiming issues.
3.	In generating the initial node labels using LLMs, it seems that only the feature information is utilized and no structure is considered. Since it is well-known in the graph machine learning literature that both features and structures greatly affect the node labels, there exists a large room for improvement.
4.	There are some missing related works regarding zero-shot node classification such as [1-2], which should be added.
5.	I also wonder how different LLMs affect the model (the reported results are all based on GPT-3.5-turbo).

[1] Zero-shot Node Classification with Decomposed Graph Prototype Network, KDD’21
[2] Dual Bidirectional Graph Convolutional Networks for Zero-shot Node Classification, KDD’22

**Questions:**

See Weaknesses above

---

> ### Author Response · Authors · 2023-11-18
> **Response to Reviewer HZKD (1/6)**
>
> Q1: Though I acknowledge that the proposed method is a valid solution, the technical contribution of the paper is somewhat limited, especially considering that the three major components are largely based on heuristic observations, and the rest are based on existing LLMs and GNNs. It would make the paper stronger if some theoretical analyses could be provided for the proposed components.
>
> **Response** Thanks for your great comments. We first summarize our responses and then provide more details. (1) We agree that theoretical analysis is important to support the effectiveness of our proposed methods. We use theoretical analysis to show why difficulty-aware selection can get high-quality annotations. We also add the following part into Appendix L. (2) We would like to strengthen our contributions: 1. We propose a flexible pipeline LLMGNN which demonstrates effectiveness with low costs; 2. We propose three effective strategies: difficulty-aware selection, confidence-aware annotations, and post-filtering to further enhance the effectiveness of LLMGNN.
>
> Q1 (1) One important heuristic used in our pipeline is difficulty-aware selection. Here, we analyze why it can enhance the annotation quality of LLMs.
>
>
>
> Given the parameter distribution of LLMs $\mathcal{Q}$, the parameter distribution of encoder $\mathcal{P}$ (SBERT), we assume ground truth $Y\_L\in \mathcal{R}^{N \times M}$, pseudo label $Y \in \mathcal{R}^{N \times M}$,  and node features encoded by the encoder $X\in \mathcal{R}^{N \times d}$. Here, $M$ denotes the number of classes and $d$ denotes the hidden dimension. Our objective is to maximize the accuracy of annotations, which is thus to minimize the discrepancies between $Y$ and $Y\_{L}$.
>
> $$
> \\min\_{(n\_{1},...,n\_{k})}{\\mathbb{E}\_{\\theta\\sim Q}{f(Y)}} = \\ell (Y,Y\_{L})=\\sum\_{i=1}^{k} \\ell({y}^{n\_{i}} , y\_{L}^{n\_{i}})
> $$
>
> where $(n\_{1},...,n\_{k})$ is the index of the $k$ selected nodes, and $f(Y)$ is defined as follows, where ${x}^{n\_{i}}$ represents the feature of node ${n\_{i}}$, while $x\_{L}^{n\_{i}}$ represents the unknown latent embedding of node ${n\_{i}}$.
> $$
> f(Y)=\\ell(Y,Y\_{L})=\\sum\_{i=1}^{k} \\ell({y}^{n\_{i}} , y\_{L}^{n\_{i}})= \\ln(1+\\sum\_{i=1}^{k}\\|{x}^{n\_{i}} - x\_{L}^{n\_{i}}\\|^{2})= \\ln(1+\\|X-X\_{L}\\|^{2})
> $$
>
>
> Since we only have access to the node features $X$ generated by encoder $\theta$, we need to make a connection between $\mathcal{Q}$ and $\mathcal{P}$.
>
> Lemma 1. For any annotation $y$ generated by LLMs, $\mathbb{E}\_{\theta\sim \mathcal{Q}}f(y)\leq \log \mathbb{E}\_{\theta\sim \mathcal{P}} \exp(f(y))+KL( \mathcal{Q} \| \mathcal{P})$
>
> Proof:
> $$
> \\begin{aligned}
> &\\\\
> &{\\mathbb{E}\_{\\theta^{\\prime}\\sim \\mathcal{P}}{f(y)}}=\\int f(y) p(y) d y=\\int f(y) \\frac{p(y)}{q(y)} q(y) d y=\\mathbb{E}\_{\\theta\\sim \\mathcal{Q}} f(y) \\frac{p(y)}{q(y)}\\\\
> &\\text{Since}\\quad KL(\\mathcal{Q} \\| \\mathcal{P})=\\mathbb{E}\_{\\theta\\sim Q} \\log \\frac{q(y)}{p(y)}\\\\
> &\\log \\mathbb{E}\_{\\theta^{\\prime}\\sim \\mathcal{P}} f(y)= \\log\\mathbb{E}\_{\\theta\\sim \\mathcal{Q}} f(y) \\frac{p(y)}{q(y)} \\geq\\mathbb{E}\_{\\theta\\sim \\mathcal{Q}}\\log f(y) \\frac{p(y)}{q(y)}=\\mathbb{E}\_{\\theta\\sim \\mathcal{Q}}\\log f(y)-KL( \\mathcal{Q} \\| \\mathcal{P}) \\\\
> \\end{aligned}
> $$
>
> Then, we transform the objective into
> $$
> \\min\_{(n\_{1},...,n\_{k})} \\mathbb{E}\_{\\theta^{\\prime}\\sim \\mathcal{P}} \\exp(f(Y))
> $$
>
> Assuming the label distribution $\\mathcal{H}$, we further have
> $$
> \\min\_{(n\_{1},...,n\_{k})} \\mathbb{E}\_{Y\\sim \\mathcal{H}} \\mathbb{E}\_{\\theta^{\\prime}\\sim \\mathcal{P}} \\exp(f(Y)) = \\min\_{(n\_{1},...,n\_{k})} \\mathbb{E}\_{Y\\sim \\mathcal{H}} \\mathbb{E}\_{\\theta\\sim \\mathcal{P}}\\|X-X\_{L}\\|^{2} \\\\
> $$
>
> Assuming $X$ and $X\_{L}$ follows gaussian distribution, where $X\sim N(\mu\_{i},\sigma\_{i});X\_L\sim N(\mu\_{j},\sigma\_{j})$, then
>
> $$
> \\begin{aligned}
> \\mathrm{E}\\left(\\|X-X\_{L}\\|^{2}\\right) & =\\mathrm{E}\\left(\\left\\|\\left(X-\\mu\_{i}\\right)-\\left(X\_{L}-\\mu\_{j}\\right)+\\left(\\mu\_{i}-\\mu\_{j}\\right)\\right\\|^{2}\\right) \\\\
> & =\\mathrm{E}\\left(\\left\\|X-\\mu\_{i}\\right\\|^{2}\\right)+\\mathrm{E}\\left(\\left\\|X\_{L}-\\mu\_{j}\\right\\|^{2}\\right)+\\left\\|\\mu\_{i}-\\mu\_{j}\\right\\|^{2} \\\\
> & =n \\sigma\_{i}^{2}+n \\sigma\_{j}^{2}+\\left\\|\\mu\_{i}-\\mu\_{j}\\right\\|^{2}
> \\end{aligned}
> $$
>
> Since $(\mu\_{j},\sigma\_{j})$ is unknown, $\mu\_{i}$ is fixed, thus we want to minimize $\sigma\_{i}$.  Given an arbitrary node, $\sigma\_{i}$ can be viewed as the distance to the clustering centers. The smaller $\sigma\_{i}$ is, the smaller the corresponding $\mathrm{E}\left(\|X-X\_{L}\|^{2}\right)$ also becomes, indicating that the minimum value of $\mathbb{E}\_{\theta\sim \mathcal{Q}}f(y)$ is attained when $\sigma\_{i}$ is at its smallest. This demonstrates why nodes closer to the clustering centers are "preferred" by LLMs and thus achieve better annotation quality.

---

> ### Author Response · Authors · 2023-11-18
> **Response to Reviewer HZKD (2/6)**
>
> Q1: Though I acknowledge that the proposed method is a valid solution, the technical contribution of the paper is somewhat limited, especially considering that the three major components are largely based on heuristic observations, and the rest are based on existing LLMs and GNNs. It would make the paper stronger if some theoretical analyses could be provided for the proposed components.
>
> Q1 (2) Besides LLMGNN we propose, we want to mention that as far as we know, we are the first to study how to improve the annotations generated by a real-world annotator. Previous works like [1] all use synthetic noisy labels, whose patterns are much simpler.
>
> [1] Zhang W, Wang Y, You Z, et al. Rim: Reliable influence-based active learning on graphs[J]. Advances in Neural Information Processing Systems, 2021, 34: 27978-27990.

---

> ### Author Response · Authors · 2023-11-18
> **Response to Reviewer HZKD (3/6)**
>
> Q2: The authors should more explicitly mention that their proposed method only works for text-attributed graphs rather than any general graph, e.g., in the abstract and introduction. Otherwise, the paper may have overclaiming issues.
>
> **Response** Thanks for your great comments. We agree that our pipelines focus more on text-attributed graphs, while our methods have the potential to be applied to more types of graphs. We add discussions on this problem in the revision.
>
> Currently, for the task of node classification, most datasets feature text-type nodes, and thus our method is well-suited for the majority of datasets and tasks. Furthermore, our approach has the potential to be extended to more graphs with different types of node features. In a recent paper [1], the authors introduced a prompt-based method that aligns all types of node features through natural language, allowing large language models to naturally process problems in the textual domain with notable success. For discrete features like bag of words, [2] also propose a potential way to let large language models handle them by converting the original features into clustering centers. These methods may be integrated with our work and extend our methods to more types of graphs. Your point is incredibly insightful and valuable, and we consider it an excellent extension of our work. In our revised version, we have included a discussion about this issue in both the abstract, introduction, and preliminaries.
>
> [1] Liu H, Feng J, Kong L, et al. One for All: Towards Training One Graph Model for All Classification Tasks[J]. arXiv preprint arXiv:2310.00149, 2023.
>
> [2] Zhao J, Zhuo L, Shen Y, et al. Graphtext: Graph reasoning in text space[J]. arXiv preprint arXiv:2310.01089, 2023.

---

> ### Author Response · Authors · 2023-11-18
> **Response to Reviewer HZKD (4/6)**
>
> Q3: In generating the initial node labels using LLMs, it seems that only the feature information is utilized and no structure is considered. Since it is well-known in the graph machine learning literature that both features and structures greatly affect the node labels, there exists a large room for improvement.
>
> **Response** Thanks for your great comments, which inspire us to have a further look at structure-aware prompts. We find that incorporating structural information in the annotation prompt **can not largely improve performance** with the current prompt design.
>
>
> We demonstrate both the accuracy of LLMs' annotations (the ratio of LLMs' annotations matching ground truth labels) and the accuracy of LLM-GNN in the following table. "No struct" means the "hybrid" (combining both "TopK" and "Most Voting") prompt strategy used in our papers. "Struct" is the new version that combines the "neighbor summarization" strategy in [1] and the "hybrid" prompt strategy.
>
> |       (Annotation)            	|         Cora    	|       Cora   	|      CiteSeer 	|     CiteSeer    	|
> |-----------------|---------------|------------|-------------|---------------|
> |                   	|      No Struct  	|      Struct  	|     No Struct 	|      Struct     	|
> |        FeatProp   	|   70.71 ± 0.00  	| 64.52 ± 0.34 	|  66.39 ± 2.83 	|  67.50  ± 0.65  	|
> |     FeatProp + PS 	|    73.81 ± 1.52 	| 63.69 ± 1.11 	|  78.12 ± 1.07 	|   78.12 ± 1.45  	|
>
> |         (LLMGNN)          	|         Cora    	|        Cora   	|      CiteSeer 	|     CiteSeer    	|
> |-----------------|---------------|-------------|-------------|---------------|
> |                   	|      No Struct  	|       Struct  	|     No Struct 	|      Struct     	|
> |        FeatProp   	|    72.82 ± 0.08 	|  69.08 ± 0.39 	| 66.61 ± 0.55  	|   65.92 ± 0.43  	|
> |     FeatProp + PS 	|    75.54 ± 0.34 	|  67.55 ± 0.70 	|  69.06 ± 0.32 	|   67.80 ± 0.45  	|
>
> We observe that:
> 1. Incorporating structural information in the annotation prompts can not improve the performance, and may affect the effectiveness of post-selection. For example, the annotation quality even drops after adding structural information.
> 2. Incorporating structural information will increase the costs of prompts since the length of prompts becomes longer.
>
> Based on these points, we think that using text-only annotation prompts for LLMs is appropriate in the current stage. However, we also admit that how incorporating structural information in LLMs is a valuable future direction.
>
> [1] Chen Z, Mao H, Li H, et al. Exploring the potential of large language models (llms) in learning on graphs[J]. arXiv preprint arXiv:2307.03393, 2023.

---

> ### Author Response · Authors · 2023-11-18
> **Response to Reviewer HZKD (5/6)**
>
> Q4: There are some missing related works regarding zero-shot node classification such as [1-2], which should be added.
>
> **Response** Thanks for your great comments which inspire us to further explore the effectiveness of our paper under the **zero-shot node classification** setting. We first summarize our responses and explain the details. (1) We've cited those related works and add a discussion section. However, **there are some differences between the setting** of label-free node classification and zero-shot node classification; (2) For zero-shot node classification, our methods can still achieve promising performance.
>
>
> (1) The setting for label-free node classification involves training a GNN using annotations generated by LLMs, without any pre-existing labels. In contrast, the zero-shot node classification setting in these two papers involves training a model with training data for training classes (for example, A, B, C), which can then be generalized to new classes in the testing phase (for example, D, E, F). In practice, our label-free methods can be used to solve zero-shot node classification by directly annotating the new classes. However, those zero-shot node classification methods can not be directly applied to label-free node classification.
>
> (2) In the following table, we compare the performance of LLMGNN to zero-shot node classification baselines DGPN [1], DBiGCN [2], and GraphCEN [3] in zero-shot node classification settings. For LLMGNN, we use the featprop + Post Selection with 20% pruning ratio and 140 annotation budget. We take the experimental settings from [3].
>
> |          	| Cora  	| CiteSeer 	|
> |----------|-------|----------|
> | DGPN     	| 33.76 	| 37.74    	|
> | DBiGCN   	| 45.08 	| 38.57    	|
> | GraphCEN 	| 48.43 	| 40.77    	|
> | LLMGNN   	| 79.45 	| 66.67    	|
>
> We can see that our methods can consistently beat all those baselines under the zero-shot node classification setting. We appreciate your comments which inspire us to demonstrate the strong capability of our methods in traditional zero-shot node classification problems.
>
> [1] Wang Z, Wang J, Guo Y, et al. Zero-shot node classification with decomposed graph prototype network[C]//Proceedings of the 27th ACM SIGKDD Conference on Knowledge Discovery & Data Mining. 2021: 1769-1779.
>
> [2] Yue Q, Liang J, Cui J, et al. Dual Bidirectional Graph Convolutional Networks for Zero-shot Node Classification[C]//Proceedings of the 28th ACM SIGKDD Conference on Knowledge Discovery and Data Mining. 2022: 2408-2417.
>
> [3] Ju W, Qin Y, Yi S, et al. Zero-shot Node Classification with Graph Contrastive Embedding Network[J]. Transactions on Machine Learning Research, 2023.

---

> ### Author Response · Authors · 2023-11-18
> **Response to Reviewer HZKD (6/6)**
>
> Q5: I also wonder how different LLMs affect the model (the reported results are all based on GPT-3.5-turbo).
>
> **Response** Thanks for your great comments. We have added experiments using other large language models like GPT-4. Surprisingly, **we find that GPT4 brings no clear improvement compared to the results of GPT-3.5-turbo.**
>
> We show the results in the following table. We can see that GPT4 only achieves clear improvements in **DA-GraphPart** on Cora. We appreciate your questions to help us have an exploration for this interesting phenomenon.
>
>
> |   (LLMGNN)   	|     Cora     	|     Cora     	|    CiteSeer    	|   CiteSeer   	|
> |------------|------------|------------|--------------|------------|
> |              	|    GPT3.5    	|     GPT4     	|     GPT3.5     	|     GPT4     	|
> |    Random    	| 70.48 ± 0.73 	| 70.03 ± 1.42 	| 65.11   ± 1.12 	| 67.00 ± 1.21 	|
> |   GraphPart  	| 69.54 ± 2.18 	| 70.65 ± 0.98 	|  66.59 ± 1.34  	| 65.66 ± 0.74 	|
> | DA-GraphPart 	| 69.64 ± 0.30 	| 75.06 ± 0.08 	|  69.88 ± 1.81  	| 69.92 ± 1.10 	|
> |   FeatProp   	| 72.82 ± 0.08 	| 69.46 ± 0.69 	|  66.61 ± 0.55  	| 66.36 ± 0.77 	|
> |  PS-FeatProp 	| 75.54 ± 0.34 	| 71.76 ± 0.33 	|  69.06 ± 0.32  	| 69.15 ± 1.06 	|
>
> One possible reason for this phenomenon is that the primary area where GPT-4 surpasses GPT-3.5 is in complex reasoning capabilities. However, in tasks such as text classification, the difference between the two is not very significant. This phenomenon has also been observed in text classification tasks in other fields, such as financial text classification [1].
>
> [1] Loukas L, Stogiannidis I, Malakasiotis P, et al. Breaking the bank with chatgpt: Few-shot text classification for finance[J]. arXiv preprint arXiv:2308.14634, 2023.

---

> > ### Comment · Reviewer_HZKD · 2023-11-20
> > **Thanks for the rebuttal**
> >
> > Thanks for the rebuttal, which I do think improve the quality of the paper. I have improved my score accordingly.
> >
> > I also have a follow-up question. In the newly provided results, it seems neither structural information nor advanced LLM can further improve the results. Does that somewhat implicate that, the task of assigning initial node labels is acturally very easy, so that GPT-3.5 with only textual information is nearly perfect  Or maybe the initial node labels themselves are not very critical, so that the results are not sensitive to them?

---

> > > ### Author Response · Authors · 2023-11-20
> > > **Response to Reviewer HZKD (1/2)**
> > >
> > > First, we want to thank you for your response and support. We are glad to know that our rebuttal has addressed your concerns.
> > >
> > > Q6: I also have a follow-up question. In the newly provided results, it seems neither structural information nor advanced LLM can further improve the results. Does that somewhat implicate that, the task of assigning initial node labels is acturally very easy, so that GPT-3.5 with only textual information is nearly perfect Or maybe the initial node labels themselves are not very critical, so that the results are not sensitive to them?
> > >
> > > **Response** Thanks for your great question. We would like to first summarize the key points and then explain the details. (1) Achieving high performance on node classification is hard for LLMs-as-predictors because of the **annotation bias** for labels. [1,2] (2) Such kind of biases can be mitigated by structural information [3]. However, current structure-aware prompts cannot help LLMs well utilize the structural information [4,5], and GNNs are currently **better** tools to utilize graph structure. We mainly focus on the pipeline design in this paper, and more powerful structure-aware prompts may further improve the effectiveness of LLMGNN. (3) The initial node label is critical to the final results.
> > >
> > >
> > > Q6-(1) Assigning correct labels for node classification is hard for LLMs. To begin with, we show the **annotation quality** (the ratio of annotations matching the ground truth labels) of GPT3.5 and GPT4 by **randomly** selecting the annotated samples ($140$ nodes for Cora, and $120$ nodes for CiteSeer, and **we control the seed to make sure we select the same set of nodes**). We can see that GPT4 doesn't give much better annotations (in terms of accuracy) than GPT3.5. The overall performance (less than 70%) indicates that node classification is not an easy task for LLMs.
> > >
> > > |   | Cora | Citeseer |
> > > |--------|--------|--------|
> > > | GPT3.5     | 67.62 ± 2.05     | 66.39 ± 8.62      |
> > > | GPT4     | 68.81 ± 1.87     | 65.56 ± 9.29     |
> > >
> > > If we further check the annotation results given by these two different models, we find that one common bottleneck preventing these models from getting better results is the annotation **bias** (also can be viewed as "label ambiguity") of the labels. We showcase one example below
> > >
> > > ```
> > > Attribute of one node from Citeseer:
> > > 'Discovering Web Access Patterns and Trends by Applying OLAP and
> > > Data Mining Technology on Web Logs As a confluence of data mining and WWW
> > > technologies, it is now possible to perform data mining on web log
> > > records collected from the Internet web page access history...
> > > ```
> > >
> > > The ground truth label for this node is "Database", while the prediction of both two LLMs is "Information Retrieval". From human beings' understanding, both "Database" and "Information retrieval" are somewhat reasonable categories of this node. However, because of the single label setting of node classification, only one of them is correct. This somehow demonstrates that to get high accuracy on node classification, the models need to capture such kinds of dataset-specific **bias** in the annotation (the information in LLMs is more like "commonsense knowledge", so there can be a mismatch). Structural information may help us find such kind of **bias**. For example, if the category of neighboring nodes is more easier to predict, and most of them should be related to "Database", then this node is more likely to come from "Database". This phenomenon has also been observed in [2].

---

> > > ### Author Response · Authors · 2023-11-20
> > > **Response to Reviewer HZKD (2/2)**
> > >
> > > Q6-(2) So, why LLMs with structure-aware prompts can not improve their performance by introducing that structural information? It's mainly because of LLMs' poor understanding capability of structural information. [3] and [4] try various kinds of structure-aware prompts, and find that LLMs present limited structure reasoning abilities with current prompt designs.  In [2] and [5], they show that incorporating **neighboring ground truth labels** can improve the performance. However, ground truth labels are not available in the label-free settings. As a comparison, we find that merely summarizing the neighboring contents may not effectively improve the performance. Under the current structure-aware prompt design, LLMs can only utilize the structural information in a very limited manner (like using neighboring labels). As a comparison, message-passing GNNs can capture complicated structural information effectively and efficiently. For example, we try replacing GNN with MLP in LLMGNN in the following table, and we observe a huge performance gap.
> > >
> > > |  | LLMs-as-Predictors | LLMGNN | LLMMLP |
> > > |--------|--------|--------|--------|
> > > | Cora     |  68.33    |  76.23    | 67.06 |
> > >
> > > Based on the above reasons, we only use textual information in the labeling process, while in the training process of GNN, we utilize structural information. At the current stage, we believe this to be a more effective paradigm. This highlights the motivation for us to design LLMGNN which can enjoy the advantages of both LLMs and GNNs while mitigating their limitations.
> > >
> > > **We also think that designing an effective structure-aware prompt is a valuable future direction**, and methods like Agent-based prompting [6] (multi-round prompt) may help to further improve the performance. The main focus of our paper is to propose a flexible framework that supports various kinds of prompt designs, and new prompt designs can also enhance the effectiveness of our framework.
> > >
> > >
> > > Q6-(3) Finally, we want to show that initial node labels are critical to the final performance. For example, if we correct a portion of incorrect annotations to the right ones, we may improve the performance of LLMGNN. The "original quality" means the original annotation given by LLMs. "+k%" means that we randomly turn "k%" wrong annotations into correct ones.
> > >
> > >
> > > |  | original quality | +7% | +12% | +17% |
> > > |--------|--------|--------|--------|--------|
> > > | Random     |  70.48    | 72.66     |  75.92       |   78.58    |
> > > | PS-FeatProp-W | 76.23   | 77.95       |  80.25      |  81.96      |
> > >
> > >
> > > Moreover, we want to show that the annotation quality will influence the scaling behavior of LLMGNN when we increase the budget. From the following table (the first row means the budget equal to $k \times \text{number of classes}$  ), we can see that (1) When the budget is low, the difference between LLMGNN trained by high-quality and low-quality annotations is smaller (<20); when the budget is high, this difference becomes larger(>20); (2) the performance of LLMGNN is closely related to the annotation quality (the performance of LLMGNN on PubMed is better than one on Cora). With better annotation quality, LLMGNN may potentially achieve a higher performance upper bound when we gradually increase the budgets.
> > >
> > > |        | 5     | 10    | 15    | 20    | 25    | 40    | 80    | 160   |
> > > |--------|-------|-------|-------|-------|-------|-------|-------|-------|
> > > | Cora   | 57.35 | 66.45 | 68.42 | 70.17 | 69.64 | 70.68 | 72.07 | 72.73 |
> > > | Pubmed | 62.49 | 64.56 | 72.44 | 73.16 | 75.92 | 77.8  | 82.38 | 82.5  |
> > >
> > >
> > > [1] Li Y, Xiong M, Hooi B. GraphCleaner: Detecting Mislabelled Samples in Popular Graph Learning Benchmarks[J]. arXiv preprint arXiv:2306.00015, 2023.
> > >
> > > [2] Chen Z, Mao H, Li H, et al. Exploring the potential of large language models (llms) in learning on graphs[J]. arXiv preprint arXiv:2307.03393, 2023.
> > >
> > > [3] Nt H, Maehara T. Revisiting graph neural networks: All we have is low-pass filters[J]. arXiv preprint arXiv:1905.09550, 2019.
> > >
> > >
> > > [4] Guo J, Du L, Liu H. GPT4Graph: Can Large Language Models Understand Graph Structured Data? An Empirical Evaluation and Benchmarking[J]. arXiv preprint arXiv:2305.15066, 2023.
> > >
> > > [5] Wang H, Feng S, He T, et al. Can Language Models Solve Graph Problems in Natural Language?[J]. arXiv preprint arXiv:2305.10037, 2023.
> > >
> > > [6] Huang J, Zhang X, Mei Q, et al. Can LLMs Effectively Leverage Graph Structural Information: When and Why[J]. arXiv preprint arXiv:2309.16595, 2023.
> > >
> > > [7] Wang L, Ma C, Feng X, et al. A survey on large language model based autonomous agents[J]. arXiv preprint arXiv:2308.11432, 2023.
> > >
> > >
> > > Please let us know in case there still remain outstanding concerns, and if so, we will be happy to respond.

---

> > > > ### Comment · Reviewer_HZKD · 2023-11-22
> > > > **Thanks for the rebuttal**
> > > >
> > > > Thanks for the clarification and I have improved my score accordingly. I encourage the authors to incorporate these discussions in the revision.

---

> > > > > ### Author Response · Authors · 2023-11-22
> > > > > **Thanks for your response and support**
> > > > >
> > > > > Thanks for your response and support. We are glad to know that our rebuttal has addressed your concerns. Please let us know in case there remain outstanding concerns, and if so, we will be happy to respond. We've uploaded a new revision that incorporates all related discussions.

---

### Official Review · Reviewer_JunD · 2023-10-30

**Soundness:** 3 good
**Presentation:** 3 good
**Contribution:** 2 fair
**Rating:** 6
**Confidence:** 4

**Summary:**

Given a text attributed graph without labels, the paper proposes a cost-effective method that combines LLMs and GNNs to annotate the labels in four steps. In the first step, the paper selects the nodes that should be annotated and terms it  as difficulty aware selection. It combines active learning (selection) techniques along with a difficulty score which is based on the distance from the center of a cluster. In the second step, annotations are created for the selected nodes using an LLM which also generates a confidence score. In the third step, nodes are pruned such that the confidence score of LLMs is high without significantly changing the diversity of nodes (via change in entropy). Finally, a GNN is trained on the graph to generate labels for other nodes.

**Strengths:**

- The idea of label-free annotation using LLMs on text attributed graphs is an interesting research direction introduced by the paper.

- The paper has experimented with different datasets and incorporated various existing techniques to come up with a cost-effective model.

- This paper appropriately balanced traditional graph active selection criteria with annotation quality by incorporating difficulty-aware active selection with post filtering to obtain training nodes from LLM.

**Weaknesses:**

- Difficulty aware (DA) selection: According to the paper, LLMs annotation quality degrades when they have to annotate nodes which are away from the centers. It implies that the LLMs annotation quality would suffer in case of the diverse nodes (away from center).  However, GNNs accuracy will only improve if the nodes are diverse. Hence, difficulty aware selection i.e. use of c-density might not always help and, in fact, it may hinder in some cases. This is also evident from the results shown in Table 2 : Active_Selection_Methods and the corresponding  DA-Active_Selection_Methods show similar performance on average across different techniques (i.e., selection methods). Moreover, for at least 50% of the cases, the DA-method (row 2) performs poorly compared to the corresponding active learning method (row 1).

- Though the accuracy from the LLM-GNN model is good (and of course, the model is efficient), it couldn't outperform LLM as a predictor (Table 3).

- The methods are heuristics and do not have theoretical evidence.

**Questions:**

- Post-filtering (PS): How useful are the confidence scores generated by the LLMs (Appendix F.1, table 6)? Showing the mean and variance of confidence score for each dataset may help in understanding its impact on performance.

- Providing the number of nodes selected in each step in the experiment will also help in understanding the effects of the steps. For instance, could you please provide the number of selected nodes during active selection and DA-active selection? Also, what is the pruning ratio when applying PS? All this information can help in understanding the efficacy of the steps.

- PS-DA-methods have not performed well in most cases compared to DA-methods (Table 2). Any insights on this can help in understanding these steps better.

Minor Questions:

- Detail explanation on f_{act}(vi) is missing

- It is mentioned in page 5 that the detailed descriptions and full prompt examples are shown in Appendix D. However, it is missing any detailed descriptions.

Typos:

- Page 7: (4) Combing - supposed to be combining?

- Page 3: GNN modelson- models on (space missing)

---

> ### Author Response · Authors · 2023-11-18
> **Response to Reviewer JunD (1/7)**
>
> W1: Difficulty aware (DA) selection: According to the paper, LLMs annotation quality degrades when they have to annotate nodes that are away from the centers. It implies that the LLMs annotation quality would suffer in the case of the diverse nodes (away from the center). However, GNNs accuracy will only improve if the nodes are diverse. Hence, difficulty aware selection i.e. use of c-density might not always help and, in fact, it may hinder in some cases. This is also evident from the results shown in Table 2 : Active\_Selection\_Methods and the corresponding DA-Active\_Selection\_Methods show similar performance on average across different techniques (i.e., selection methods). Moreover, for at least 50\% of the cases, the DA-method (row 2) performs poorly compared to the corresponding active learning method (row 1).
>
> **Response** Thanks for your great comments. We want to first summarize our responses and then explain them in details. Applying the original DA may lead to clear performance degradation, which is due to the class imbalance problem by a small clustering center number $K$ on datasets like Pubmed. **This problem can be solved** by increasing $K$ to balance the diversity and annotation quality.
>
> The reason for DA's bad performance on Pubmed is when performing K-means clustering with $C-Density$, the number of clustering centers $K$ is fixed as the number of classes. This approach can achieve good labeling quality in most cases. On datasets like Pubmed where the number of classes is small (only 3), the points selected by $C-Density$ tend to cause class imbalance (the annotation quality is still very good). To solve this problem, we find that by setting a larger value for the number of clustering centers $K$, the new $ C-Density^{\prime}$ obtained can effectively solve this problem. We show the results in the following table. $k$ means the number of clustering centers to calculate $C-Density^{\prime}$, $C$ means the number of classes, and budgets means the number of selected annotated nodes. We use AGE as a demonstration to show the idea. From the results, we can see that setting a larger $K$ can improve the performance of DA on Pubmed.
>
> |                  	| 	 Pubmed 	|
> |----------------|------|
> |        AGE       	|  74.55 	|
> |    DA-AGE(k=C)   	|   55.36 	|
> |   DA-AGE(k=2C)   	|   57.38 	|
> | DA-AGE(k=budget) 	|  75.71 	|
>
>
> In the following table, we demonstrate that DA can improve the performance of diversity-aware selection methods most of the time.
>
> |              	|     Cora     	|   CiteSeer   	|    Pubmed    	|    WikiCS    	|
> |------------|------------|------------|------------|------------|
> |      AGE	     	| 69.15 ± 0.38 	| 54.25 ± 0.31 	| 74.55 ± 0.54 	| 55.51 ± 0.12 	|
> |    DA-AGE    	| 74.38 ± 0.24 	| 59.92 ± 0.42 	| 74.20 ± 0.51 	| 59.39 ± 0.21 	|
> |      RIM     	| 69.86 ± 0.38 	| 63.44 ± 0.42 	| 76.22 ± 0.16 	|  66.72± 0.16 	|
> |    DA-RIM    	| 73.99 ± 0.44 	| 60.33 ± 0.40 	| 79.17 ± 0.11 	|  67.82± 0.32 	|
> |   GraphPart  	| 68.57 ± 2.18 	| 66.59 ± 1.34 	| 77.50 ± 1.23 	| 67.28 ± 0.87 	|
> | DA-GraphPart 	| 69.35 ± 1.92 	| 69.37 ± 1.27 	| 79.49 ± 0.85 	| 68.72 ± 1.01 	|
>
> We can see for 10/12 cases, DA significantly improves the performance. In other cases, it also achieves comparable performance.

---

> ### Author Response · Authors · 2023-11-18
> **Response to Reviewer JunD (2/7)**
>
> W2: Though the accuracy from the LLM-GNN model is good (and of course, the model is efficient), it couldn't outperform LLM as a predictor (Table 3).
>
> **Response** Thanks for your great comments. Despite the superior performance of LLMs-as-Predictors in some datasets, it's not practical to be deployed in real-world scenarios because of **efficiency**. Our methods can **scale to large-scale graphs with promising performance**.
>
>  Although the original LLMs-as-predictors can achieve good performance, it's not practical to be applied to even medium-scale datasets like Arxiv. In the original paper [1], it's only applied to a small subset of the datasets to evaluate the performance. Taking Arxiv as an example, considering OpenAI's current limit of 60 accesses per minute, completing all predictions on Arxiv would take 44 hours, whereas LLMGNN only requires a few minutes. Our methods extend the original pipeline to real-world large-scale datasets.
>
> Then, we compare the performance and the economic costs of these two pipelines to further demonstrate the effectiveness of our methods in the following two tables. The cost is estimated in units of dollars. On datasets like Citeseer, and Products, the gap between LLMGNN and LLMs-as-predictors is very small. On Cora, LLMGNN can even outperform LLMs-as-predictors. Only on PubMed, there's a relatively large gap between LLM-GNN and PubMed. It's because of LLMs' superior performance on this dataset, which is related to the shortcut prediction in attribute [1]. When LLMs' performance is superior, we can also add more budgets to further improve the performance of LLMGNN. ***For example, if we raise the budget in PubMed from 0.05 to 0.25, we can raise the accuracy of LLMGNN from 74.98 to 82.***
>
>
>
> | Performance        	| Cora  	| CiteSeer 	| PubMed 	| WikiCS 	| Arxiv 	| Products 	|
> |--------------------|-------|----------|--------|--------|-------|----------|
> | LLMGNN             	| 75.54 	| 69.06    	| 74.98  	| 66.09  	| 66.14 	| 74.91    	|
> | LLMs-as-Predictors 	| 67.33 	| 66.33    	| 87.33  	| 71     	| 73.67 	| 75.33    	|
>
> | Costs              	| Cora  	| CiteSeer 	| PubMed 	| WikiCS 	| Arxiv 	| Products 	|
> |--------------------|------|----------|--------|--------|-------|----------|
> | LLMGNN             	| 0.11  	| 0.1      	| 0.05   	| 0.16   	| 0.63  	| 0.74     	|
> | LLMs-as-Predictors 	| 1.26  	| 1.5      	| 9.2    	| 5.46   	| 79    	| 1952     	|
>
> In summary, the economic costs and efficiency of our methods are consistently much lower than LLMs-as-predictors. It's a valuable future direction to study how to further improve the performance of LLMGNN.
>
>
> [1] Chen Z, Mao H, Li H, et al. Exploring the potential of large language models (llms) in learning on graphs[J]. arXiv preprint arXiv:2307.03393, 2023.

---

> ### Author Response · Authors · 2023-11-18
> **Response to Reviewer JunD (3/7)**
>
> W3: The methods are heuristics and do not have theoretical evidence.
>
> **Response** Thanks for your great comments. We agree that theoretical analysis is important to support the effectiveness of our proposed methods. We show why Difficulty-aware selection can improve the annotation quality of LLMs. We also add the following part into Appendix L.
>
> Given the parameter distribution of LLMs $\mathcal{Q}$, the parameter distribution of encoder $\mathcal{P}$ (SBERT), we assume ground truth $Y_L\in \mathcal{R}^{N \times M}$, pseudo label $Y \in \mathcal{R}^{N \times M}$,  and node features encoded by the encoder $X\in \mathcal{R}^{N \times d}$. Here, $M$ denotes the number of classes and $d$ denotes the hidden dimension. Our objective is to maximize the accuracy of annotations, which is thus to minimize the discrepancies between $Y$ and $Y_{L}$.
>
> $$
> \\min_{(n_{1},...,n_{k})} {\mathbb{E}_{\theta\sim Q}{f(Y)}} = \\ell(Y,Y\_{L})=\sum\_{i=1}^{k} \\ell({y}^{n\_{i}} , y\_{L}^{n\_{i}})
> $$
>
> where $(n_{1},...,n_{k})$ is the index of the $k$ selected nodes, and $f(Y)$ is defined as follows, where ${x}^{n_{i}}$ represents the feature of node ${n_{i}}$, while $x_{L}^{n_{i}}$ represents the unknown latent embedding of node ${n_{i}}$.
> $$
> f(Y)=\\ell(Y,Y_{L})=\\sum_{i=1}^{k} \\ell({y}^{n_{i}} , y_{L}^{n_{i}})= \\ln(1+\\sum_{i=1}^{k}\\|{x}^{n_{i}} - x_{L}^{n_{i}}\\|^{2})= \\ln(1+\\|X-X_{L}\\|^{2})
> $$
>
>
> Since we only have access to the node features $X$ generated by encoder $\theta$, we need to make a connection between $\mathcal{Q}$ and $\mathcal{P}$.
>
> Lemma 1. For any annotation $y$ generated by LLMs, $\\mathbb{E}\_{\\theta\\sim \\mathcal{Q}}f(y)\\leq \\log \mathbb{E}_{\\theta\\sim \\mathcal{P}} \\exp(f(y))+KL( \\mathcal{Q} \\| \\mathcal{P})$
>
> Proof:
> $$
> \\begin{aligned}
> &\\\\
> &{\\mathbb{E}\_{\\theta^{\\prime}\\sim \\mathcal{P}}{f(y)}}=\\int f(y) p(y) d y=\\int f(y) \\frac{p(y)}{q(y)} q(y) d y=\\mathbb{E}\_{\\theta\\sim \\mathcal{Q}} f(y) \\frac{p(y)}{q(y)}\\\\
> &\\text{Since}\\quad KL(\\mathcal{Q} \\| \\mathcal{P})=\\mathbb{E}\_{\\theta\\sim Q} \\log \\frac{q(y)}{p(y)}\\\\
> &\\log \\mathbb{E}\_{\\theta^{\\prime}\\sim \\mathcal{P}} f(y)= \\log\\mathbb{E}\_{\\theta\\sim \\mathcal{Q}} f(y) \\frac{p(y)}{q(y)} \\geq\\mathbb{E}\_{\\theta\\sim \\mathcal{Q}}\\log f(y) \\frac{p(y)}{q(y)}=\\mathbb{E}\_{\\theta\\sim \\mathcal{Q}}\\log f(y)-KL( \\mathcal{Q} \\| \\mathcal{P}) \\\\
> \\end{aligned}
> $$
>
> Then, we transform the objective into
> $$
> \\min\_{(n_{1},...,n_{k})} \\mathbb{E}\_{\\theta^{\\prime}\\sim \\mathcal{P}} \\exp(f(Y))
> $$
>
> Assuming the label distribution $\\mathcal{H}$, we further have
> $$
> \\min\_{(n_{1},...,n_{k})} \\mathbb{E}\_{Y\\sim \\mathcal{H}} \\mathbb{E}\_{\\theta^{\\prime}\\sim \\mathcal{P}} \\exp(f(Y)) = \\min\_{(n_{1},...,n_{k})} \\mathbb{E}\_{Y\\sim \\mathcal{H}} \\mathbb{E}\_{\\theta\\sim \\mathcal{P}}\\|X-X_{L}\\|^{2} \\\\
> $$
>
> Assuming $X$ and $X_{L}$ follows gaussian distribution, where $X\sim N(\mu_{i},\sigma_{i});X_L\sim N(\mu_{j},\sigma_{j})$, then
>
> $$
> \\begin{aligned}
> \\mathrm{E}\\left(\\|X-X_{L}\\|^{2}\\right) & =\\mathrm{E}\\left(\\left\\|\\left(X-\\mu_{i}\\right)-\\left(X_{L}-\\mu_{j}\\right)+\\left(\\mu_{i}-\\mu_{j}\\right)\\right\\|^{2}\\right) \\\\
> & =\\mathrm{E}\\left(\\left\\|X-\\mu_{i}\\right\\|^{2}\\right)+\\mathrm{E}\\left(\\left\\|X_{L}-\\mu_{j}\\right\\|^{2}\\right)+\\left\\|\\mu_{i}-\\mu_{j}\\right\\|^{2} \\\\
> & =n \\sigma_{i}^{2}+n \\sigma_{j}^{2}+\\left\\|\\mu_{i}-\\mu_{j}\\right\\|^{2}
> \\end{aligned}
> $$
>
> Since $(\mu_{j},\sigma_{j})$ is unknown, $\mu_{i}$ is fixed, thus we want to minimize $\sigma_{i}$.  Given an arbitrary node, $\sigma_{i}$ can be viewed as the distance to the clustering centers. The smaller $\sigma_{i}$ is, the smaller the corresponding $\mathrm{E}\left(\|X-X_{L}\|^{2}\right)$ also becomes, indicating that the minimum value of $ \\mathbb{E}\_{ \\theta \\sim \\mathcal{Q}} f(y)$ is attained when $\\sigma_{i}$ is at its smallest. This demonstrates why nodes closer to the clustering centers are "preferred" by LLMs and thus achieve better annotation quality.

---

> ### Author Response · Authors · 2023-11-18
> **Response to Reviewer JunD (4/7)**
>
> Q1: Post-filtering (PS): How useful are the confidence scores generated by the LLMs (Appendix F.1, table 6)? Showing the mean and variance of the confidence score for each dataset may help in understanding its impact on performance.
>
> **Response** Thanks for your great comments. We show the mean and variance of confidence scores for each dataset in the following table and also put a figure for the Cora dataset in the revision (Appendix H). **The confidence generated by LLMs is well calibrated and can be used to reflect the reliability of annotations.**
>
> The following table demonstrates the Mean, Variance, Mode, Maximum, and Minimum of the confidence for each dataset.
> |          	|  Cora 	| CiteSeer 	| PubMed 	| WikiCS 	| Arxiv 	| Products 	|
> |--------|-----|--------|------|------|-----|--------|
> |   Mean   	| 0.764 	|   0.773  	|  0.866 	|   0.8  	|  0.89 	|   0.833  	|
> | Variance 	| 0.019 	|   0.014  	|  0.006 	|  0.012 	| 0.011 	|   0.006  	|
> |   Mode   	|  0.9  	|    0.8   	|   0.9  	|   0.8  	|  0.95 	|    0.9   	|
> |  Maximum 	| 0.967 	|   0.958  	|  0.983 	|  0.942 	| 0.992 	|   0.942  	|
> |  Minimum 	|  0.2  	|   0.233  	|  0.483 	|  0.125 	|  0.3  	|   0.317  	|
>
> For the confidence calibration plot, we find that the hybrid prompt we use can generate accurate and diverse confidence scores, which demonstrates effectiveness.

---

> ### Author Response · Authors · 2023-11-18
> **Response to Reviewer JunD (5/7)**
>
> Q2: Providing the number of nodes selected in each step in the experiment will also help in understanding the effects of the steps. For instance, could you please provide the number of selected nodes during active selection and DA-active selection? Also, what is the pruning ratio when applying PS? All this information can help in understanding the efficacy of the steps.
>
> **Response** Thanks for your great comments. For the number of nodes selected during active selection and DA-active selection, we adopt the same number of samples for each dataset as follows: the number of selected nodes is equal to the number of classes multiplied by $20$. For example, for Arxiv with $40$ classes, we select a total of $800$ nodes. For the pruning ratio, we drop 20\% of the selected nodes for every dataset except Pubmed. For Pubmed, we drop 10\% of the selected nodes because LLMs-as-predictors can achieve high accuracy on this dataset.

---

> ### Author Response · Authors · 2023-11-18
> **Response to Reviewer JunD (6/7)**
>
> Q3: PS-DA-methods have not performed well in most cases compared to DA-methods (Table 2). Any insights on this can help in understanding these steps better.
>
> **Response** Thanks for your great comments. For the effectiveness of applying PS and DA together, we agree that directly combining them without tuning can't outperform DA in most cases. If we tune the hyper-parameters, we can get good performance. However, PS-DA introduces numerous hyper-parameters which makes tuning complicated. We further show using LLMs' confidence as the weight for loss function is a better way to be combined with DA methods.
> * The reason that directly applying PS together with DA could lead to a performance decrease is that we don't tune the parameters. If we do not tune the hyperparameters, in some cases where DA has already selected a good candidate set, further adopting PS to remove some nodes will degrade the performance.
> * If we can tune the parameters, both DA and PS can be viewed as a special case of the more general PS-DA. However, in practice, PS-DA introduces many hyper-parameters, which makes it infeasible to tune the parameters without a validation set.
> * Instead of combining PS and DA in a "hard" way by filtering nodes with LLMs' confidence, we find that a "soft" way which uses the confidence as the weight for loss function works better. In the following table, we show that compared to PS-DA, combining DA with weighted loss can effectively enhance the performance of DA. Compared to PS-DA, our new approach shares the same philosophy. PS-DA can be viewed as a hard selection process while weighted cross-entropy loss can be viewed as a soft selection. We find that such kind of soft selection can effectively solve the problem of hyper-parameter tuning. The full table can be viewed in revision. "-W" means the weighted loss. We find that using a weighted loss combination has two advantages: (1) no need to tune the hyper-parameters like PS-DA; and (2) better performance.
>
>
> |           	|     Cora     	|   CiteSeer   	|
> |---------|------------|------------|
> |    AGE    	| 69.15 ± 0.38 	| 54.25 ± 0.31 	|
> |   DA-AGE  	| 74.38 ± 0.24 	| 59.92 ± 0.42 	|
> |  DA-AGE-W 	| 74.96 ± 0.22 	| 58.41 ± 0.45 	|
> | PS-DA-AGE 	| 71.53 ± 0.19 	| 56.38 ± 0.14 	|
> |    RIM    	| 69.86 ± 0.38 	| 63.44 ± 0.42 	|
> |   DA-RIM  	| 73.99 ± 0.44 	| 60.33 ± 0.40 	|
> |  DA-RIM-W 	| 74.73 ± 0.41 	| 60.80 ± 0.57 	|
> | PS-DA-RIM 	| 72.34 ± 0.19 	| 60.33 ± 0.40 	|

---

> ### Author Response · Authors · 2023-11-18
> **Response to Reviewer JunD (7/7)**
>
> Q4: Minor Questions: 1. Detail explanation on $f_{act}(vi)$ is missing; 2. It is mentioned in page 5 that the detailed descriptions and full prompt examples are shown in Appendix D. However, it is missing any detailed descriptions.
>
> **Response** Thanks for your great comments. $f_{act}(vi)$ can be any traditional graph active learning score functions, and we add a detailed explanation in the revision. Furthermore, we add more examples of the prompt and detailed descriptions in the appendix.
>
> Q5: Typos: Page 7: (4) Combing - supposed to be combining?; Page 3: GNN modelson- models on (space missing)
>
> **Response** Thanks for your great comments. We've fixed these typos in the revision.

---

> > ### Comment · Reviewer_JunD · 2023-11-18
> > **Thanks for the rebuttal!**
> >
> > Thank you for your effort on the rebuttal.
> > Based on the quality of the paper and the clarifications in the rebuttal, I am raising my score.

---

> > > ### Author Response · Authors · 2023-11-18
> > > **Thanks for your response**
> > >
> > > Thanks for your response and support. We are glad to know that our rebuttal has addressed your concerns. Please let us know in case there still remain outstanding concerns, and if so, we will be happy to respond.

---

### Official Review · Reviewer_ugc7 · 2023-10-31

**Soundness:** 2 fair
**Presentation:** 3 good
**Contribution:** 2 fair
**Rating:** 6
**Confidence:** 4

**Summary:**

This paper aims to combine Large Language Models(LLMs) and Graph Neural Networks(GNNs), leveraging their strengths. GNNs achieve promising performance when dealing with graph-structured data, while it needs abundant high-quality labels to ensure the performance. On the other hand, LLMs shows impressive zero-shot proficiency on text-attributed graphs, but it suffers from high inference costs and processing structural data. This paper suggests LLM-GNN, using LLM for annotation, providing training signals on GNN for further prediction. Moreover, the authors propose node selection strategy and confidence-aware annotation for efficient learning with high quality annotations.

**Strengths:**

- The proposed method is well-motivated.
- Each component supports motivation reasonably.
- Well written paper, it is easy to follow.

**Weaknesses:**

- The performance gain is incremental, especially for Difficulty-aware active node selection (DA).
- Explanations about experiments are not enough, and some parts are unclear. Please refer to questions for details.

**Questions:**

- In Figure 2 or Figure 12, the trend of decreasing accuracy as the distance between nodes and cluster centers increases seems somewhat weak in average accuracy. Even in Table 2, when DA is added to traditional graph active selection and when DA is added to the use of PS, there are many cases where performance actually decreases. I acknowledge that tuning was not performed, but there are still too many cases where performance declines. While the authors said that grid search would improve the performance in a specific case, it seems necessary to perform more tuning across a broader range of cases to clearly demonstrate the effectiveness of DA.
- In Table 6, is it realistic to use labels in 1-shot example when we are considering the "Label-free" setting? Furthermore, how are the confidence scores determined in that example? More detailed explanation is required.
- In Figure 4, why do some methods show an increase in performance after a budget of 70, followed by a sharp decline? Could this be attributed to class-imbalance issues during the selection process?
- While there is an example in Figure 5, it would be beneficial to demonstrate on different datasets that LLM-GNN achieves competitive performance with GNNs trained on ground truth labeled data, while significantly reducing costs.

---

> ### Author Response · Authors · 2023-11-18
> **Response to Reviewer ugc7 (1/7)**
>
> Q1-1: In Figure 2 or Figure 12, the trend of decreasing accuracy as the distance between nodes and cluster centers increases seems somewhat weak in average accuracy.
>
>
> R1-1: Thanks for your great question which inspires us to explain the details of our paper. There are line plots and bar plots in the figure. The changing trend is reflected by the line plot, which **shows a clear declining trend** while the bar plot only provides auxiliary information and may not affect the conclusion.
>
> For Figure 2 and Figure 12, **the bar** shows the average values within each region, while the blue line represents the overall mean accuracy across different regions. Although the accuracy inside each bar doesn't show a clear trend, the line plot shows a clear decreasing trend across different regions. The nodes to be annotated are selected based on the accuracy across different groups (the blue line) rather than the accuracy inside each group (the bar). As a result, **the declining trend is clear** and difficulty-aware selection can effectively select those nodes with high-quality annotations.

---

> ### Author Response · Authors · 2023-11-18
> **Response to Reviewer ugc7 (2/7)**
>
> Q1-2: Even in Table 2, when DA is added to traditional graph active selection and when DA is added to the use of PS, there are many cases where performance actually decreases. I acknowledge that tuning was not performed, but there are still too many cases where performance declines. While the authors said that grid search would improve the performance in a specific case, it seems necessary to perform more tuning across a broader range of cases to clearly demonstrate the effectiveness of DA.
>
> R1-2: Thanks for your great question. We first summarize our responses and then provide more details for the responses.
> (1) Applying the original DA may lead to clear performance degrade, which is due to the class imbalance problem by a small clustering center number $K$ on datasets like Pubmed. **This problem can be solved** by increasing $K$ to balance the diversity and annotation quality. (2) The original way to apply PS and DA together can not outperform DA. Although hyper-parameter tuning can definitely help, we agree that hyper-parameter tuning could not be flexible in practice. Therefore to further explore the potential of combining DA and PS, **we introduce a simple way to utilize the confidence as weight for loss functions and we find that the new variants can outperform DA in most cases without any hyper-parameter tuning.**
>
>
> R1-2 (1) For the effectiveness of DA, we admit that sometimes DA will degrade the performance, especially on the Pubmed dataset. The reason for this phenomenon is that when performing K-means clustering with $C-Density$, the number of clustering centers $K$ is fixed as the number of classes for labels. This approach can achieve good labeling quality in most cases. On datasets like Pubmed where the number of classes is small (only 3), the points selected by $C-Density$ tend to cause class imbalance (the annotation quality is still very good). To solve this problem, we find that by setting a larger value for the number of clustering center $K$, the new $C-Density^{\prime}$ obtained can effectively solve this problem. We show the results in the following table. $k$ means the number of clustering centers to caluclate $C-Density^{\prime}$, $C$ means the number of classes, budgets means the number of selected annotated nodes. We use AGE as a demonstration to show the idea. From the results, we can see that setting a larger $K$ can improve the performance of DA on Pubmed.
>
> |                  	| Pubmed |
> |----------------|------|
> |        AGE       	|  74.55 	|
> |    DA-AGE(k=C)   	|  55.36 	|
> |   DA-AGE(k=2C)   	|  57.38 	|
> | DA-AGE(k=budget)|  75.71 	|

---

> ### Author Response · Authors · 2023-11-18
> **Response to Reviewer ugc7 (3/7)**
>
> R1-2 (2) For the effectiveness of applying PS and DA together, we agree that directly combining them without tuning can't outperform DA in most cases. While hyper-parameter tuning can help, we also explore using LLMs' confidence as the weight for weighted cross-entropy loss that can effectively combine DA with the information generated by LLMs. we will first explain why PS-DA doesn't work.
> * The reason that directly applying PS together with DA could lead to a performance decrease is that we don't tune the parameters. If we do not tune the hyperparameters, in some cases where DA has already selected a good candidate set, further adopting PS to remove some nodes will degrade the performance.
> * In our experiments, the purpose of demonstrating PS-DA is to compare it with the use of PS and DA individually and to explain that without hyperparameter tuning on a validation set, using just one of PS or DA is a better choice. Among them, PS often achieves better quality because it can utilize the confidence information from LLMs, but it may lose some sample points due to filtering. Since DA does not need to use the confidence information from LLMs, it can employ a regular zero-shot prompt instead of a consistency prompt, making it relatively lower in cost. Choosing between DA and PS can be seen as a trade-off in terms of cost and the number of samples. DA and PS can surpass the original baseline in most cases.
> * In the following table, we show that compared to PS-DA, combining DA with weighted loss can effectively enhance the performance of DA. Compared to PS-DA, our new approach shares the same philosophy. PS-DA can be viewed as a hard selection process while weighted cross-entropy loss can be viewed as a soft selection. We find that such kind of soft selection can effectively solve the problem of hyper-parameter tuning. The full table can be viewed in revision. "-W" means the weighted loss. We find that using a weighted loss combination has two advantages: (1) no need to tune the hyper-parameters like PS-DA; and (2) better performance.
>
>
> |           	|     Cora     	|   CiteSeer   	|
> | --------- | ------------ |------------|
> |    AGE    	| 69.15 ± 0.38 	| 54.25 ± 0.31 	|
> |   DA-AGE  	| 74.38 ± 0.24 	| 59.92 ± 0.42 	|
> |  DA-AGE-W 	| 74.96 ± 0.22 	| 58.41 ± 0.45 	|
> | PS-DA-AGE 	| 71.53 ± 0.19 	| 56.38 ± 0.14 	|
> |    RIM    	| 69.86 ± 0.38 	| 63.44 ± 0.42 	|
> |   DA-RIM  	| 73.99 ± 0.44 	| 60.33 ± 0.40 	|
> |  DA-RIM-W 	| 74.73 ± 0.41 	| 60.80 ± 0.57 	|
> | PS-DA-RIM 	| 72.34 ± 0.19 	| 60.33 ± 0.40 	|
>
> In the following table, we demonstrate that DA can improve the performance of diversity-aware selection methods most of the time.
>
> |              	|     Cora     	|   CiteSeer   	|    Pubmed    	|    WikiCS    	|
> |------------|------------|------------|------------|------------|
> |      AGE	     	| 69.15 ± 0.38 	| 54.25 ± 0.31 	| 74.55 ± 0.54 	| 55.51 ± 0.12 	|
> |    DA-AGE    	| 74.38 ± 0.24 	| 59.92 ± 0.42 	| 74.20 ± 0.51 	| 59.39 ± 0.21 	|
> |      RIM     	| 69.86 ± 0.38 	| 63.44 ± 0.42 	| 76.22 ± 0.16 	|  66.72± 0.16 	|
> |    DA-RIM    	| 73.99 ± 0.44 	| 60.33 ± 0.40 	| 79.17 ± 0.11 	|  67.82± 0.32 	|
> |   GraphPart  	| 68.57 ± 2.18 	| 66.59 ± 1.34 	| 77.50 ± 1.23 	| 67.28 ± 0.87 	|
> | DA-GraphPart 	| 69.35 ± 1.92 	| 69.37 ± 1.27 	| 79.49 ± 0.85 	| 68.72 ± 1.01 	|
>
> We can see for 10/12 cases, DA leads to a significant gain. In other cases, it can also achieve comparable performance.

---

> ### Author Response · Authors · 2023-11-18
> **Response to Reviewer ugc7 (4/7)**
>
> Q2: In Table 6, is it realistic to use labels in 1-shot example when we are considering the "Label-free" setting? Furthermore, how are the confidence scores determined in that example? More detailed explanation is required.
>
> **Response** Thanks for your great question. We agree that the original version may lead to confusion since we do not clarify that we only use "1-shot" prompts to demonstrate there's no clear gap between "zero-shot" and "1-shot"  and **we use the "zero-shot" prompt in all experiments** of Section 4.
>
> In the revision, we have updated the prompt example of zero-shot case in the appendix E. We agree that "1-shot" may be not appropriate for the "label-free" setting. However, we only use 1-shot prompts in Table 1 to demonstrate that there's no clear gap between the performance of zero-shot and 1-shot prompts. For the experiment part, we only focus on zero-shot prompts.
> For the generation of multiple labels and their corresponding confidence scores in the 1-shot case, we leverage LLMs to automatically generate both the topK predictions and confidence scores according to the philosophy of [1]. We've revised the corresponding part of the paper.
>
> [1] Zhang Z, Zhang A, Li M, et al. Automatic chain of thought prompting in large language models[J]. arXiv preprint arXiv:2210.03493, 2022.

---

> ### Author Response · Authors · 2023-11-18
> **Response to Reviewer ugc7 (5/7)**
>
> Q3: In Figure 4, why do some methods show an increase in performance after a budget of 70, followed by a sharp decline? Could this be attributed to class imbalance issues during the selection process?
>
> **Response** Thanks for your great comment which points out an important phenomenon in our experiments. We first summarize our responses and then provide more details for the responses. (1) For ground truth labels, increasing the budget can usually improve the performance. However, for noisy labels, it's reasonable that increasing the budget will lead to a performance drop since labels are noisy [1]. (2) The original scales of the x-axis and y-axis are inappropriate, making the declining phenomenon appear very pronounced, when in fact **the magnitude of the decline is relatively small**. (3) The main reason for the performance degradation is attributed to the **decline in annotation quality**, rather than the class imbalance.
>
> Q3 (1) Since we use LLMs as the annotators, two factors affect the final performance of LLMGNN: **number of annotated samples**, and **annotation quality**. For clean labels, usually, the performance will gradually increase with more budgets. However, for noisy labels, it's reasonable that performance drops with more labels since the overall annotation quality degrades [1].
>
> Q3 (2) In the original figure, we use the x ticks [70, 280, 1120, 2240] with non-equal distance between every tick. This unreasonable design results in a visually pronounced appearance of the decline. We have redrawn the chart in the revision, and the corresponding results are presented in the table below, showing that the decline's magnitude is not particularly significant.
>
>
> |                            	|   35  	|   70  	|  105  	|  140  	|  175  	|  280  	|  560  	|  1120 	|
> |--------------------------|-----|-----|-----|-----|-----|-----|-----|-----|
> |         Random (CE)        	| 57.35 	| 66.45 	| 68.42 	| 70.17 	| 69.64 	| 70.68 	| 72.07 	| 72.73 	|
> |         Random(WE)         	|  59.3 	| 68.88 	| 68.77 	| 71.85 	| 71.77 	| 71.75 	| 72.81 	| 72.92 	|
> |          PS-Random         	|  55.6 	|  66.7 	| 69.86 	| 71.49 	| 71.07 	| 72.87 	| 73.81 	| 72.19 	|
> |          FeatProp          	| 65.36 	| 69.18 	| 74.07 	| 72.82 	| 72.96 	| 70.63 	| 72.53 	| 73.74 	|
> |         PS-FeatProp        	| 68.25 	|  69.8 	| 76.03 	| 75.54 	| 76.83 	| 74.11 	| 72.26 	| 72.52 	|
>
> [1] Song H, Kim M, Park D, et al. Learning from noisy labels with deep neural networks: A survey[J]. IEEE Transactions on Neural Networks and Learning Systems, 2022.

---

> ### Author Response · Authors · 2023-11-18
> **Response to Reviewer ugc7 (6/7)**
>
> Q3: In Figure 4, why do some methods show an increase in performance after a budget of 70, followed by a sharp decline? Could this be attributed to class imbalance issues during the selection process?
>
> Q3 (3) The performance degradation is mainly due to the annotation quality drop with more budgets. For random selection, the drop is not obvious because of random sampling. For active selection like PS-Featprop, the annotation quality drops with more budgets since we fixed the dropping ratio to 20\%. With more budgets, there will be correspondingly more wrong annotations.
>
>
>
> In the following table, we demonstrate the annotation quality and entropy given by PS-Featprop. Entropy can be used to measure the imbalance of selected labels.
>
> |                    	|  35  	|   70  	|  105  	|  140  	|  175  	|  280  	|  560  	|  1120 	|
> |------------------|----|-----|-----|-----|-----|-----|-----|-----|
> |       Entropy      	| 2.42 	|  2.42 	|  2.48 	|  2.55 	|  2.51 	|  2.54 	|  2.53 	|  2.56 	|
> | Annotation quality 	|  75  	| 76.79 	| 80.95 	| 75.27 	| 79.29 	| 78.12 	| 73.21 	| 71.99 	|
> |         PS-FeatProp        	| 68.25 	|  69.8 	| 76.03 	| 75.54 	| 76.83 	| 74.11 	| 72.26 	| 72.52 	|
>
> We can see that the performance drop is more correlated with the annotation quality drop. To mitigate this problem, one possible solution is to adaptively increase the dropping rate of PS-Featprop, and we may increase the performance when there's more noisy labels. This method can only mitigate the issue rather than resolve it completely. We acknowledge that exploring how to further enhance the performance of LLMGNN under high labeling rates is a very meaningful direction for future work.
>
> |                            	|   35  	|   70  	|  105  	|  140  	|  175  	|  280  	|  560  	|  1120 	|
> |--------------------------|-----|-----|-----|-----|-----|-----|-----|-----|
> |         PS-FeatProp        	| 68.25 	|  69.8 	| 76.03 	| 75.54 	| 76.83 	| 74.11 	| 72.26 	| 72.52 	|
> | PS-FeatProp(adaptive drop) 	| 68.25 	|  69.8 	| 76.03 	| 75.54 	| 76.83 	| 75.08 	| 73.54 	| 74.45 	|
>
> Another potential solution is to further improve the zero-shot performance of LLMs, like using more complex prompts or agents-related techniques [2]. Using Pubmed as an example, since LLMs can achieve near 90% accuracy on this dataset, increasing the budget can effectively improve the performance of LLMGNN. This shows that if LLMs can get good annotation quality, directly using randomly selected annotations can already get good performance.
>
> |   Pubmed  	|   15  	|   30  	|   45  	|   60  	|   75  	|  120  	|  240  	|  480  	|
> |---------|-----|-----|-----|-----|-----|-----|-----|-----|
> |   Random  	| 62.49 	| 64.56 	| 72.44 	| 73.16 	| 75.92 	|  77.8 	| 82.38 	|  82.5 	|
> | PS-Random 	| 62.84 	| 63.28 	| 72.24 	| 73.13 	| 75.49 	| 79.06 	|  82.8 	| 81.81 	|
> |  FeatProp 	| 54.67 	| 72.02 	|  75.2 	| 76.47 	| 74.39 	| 79.76 	| 81.49 	| 82.86 	|
>
> ||  Cora 	| Pubmed 	|
> |------|-----|------|
> |LLMs-as-Predictors| 68.33 	|  87.33 	|
>
> [2] Xi Z, Chen W, Guo X, et al. The rise and potential of large language model based agents: A survey[J]. arXiv preprint arXiv:2309.07864, 2023.

---

> ### Author Response · Authors · 2023-11-18
> **Response to Reviewer ugc7 (7/7)**
>
> Q4: While there is an example in Figure 5, it would be beneficial to demonstrate on different datasets that LLM-GNN achieves competitive performance with GNNs trained on ground truth labeled data, while significantly reducing costs.
>
> **Response** Thanks for your great comments which help us to refine the details of our paper.  Compared to traditional GNN and LLMs-as-Predictors, **our methods can achieve promising performance with much lower costs.**
>
> We demonstrate the economic comparison among LLM-GNN, LLMs-as-Predictors, and traditional pipelines to show the effectiveness. The first table demonstrates the performance of each pipeline. For LLMGNN and GNN, we deliberately let them produce a similar performance to conduct a fair cost comparison. For Pubmed*, we demonstrate the performance of LLMGNN with a larger budget since the performance of LLMs-as-Predictors is superior on this dataset, and increasing a little budget can improve the performance a lot.
> |     (Accuracy)     	|  Cora 	| CiteSeer 	| Pubmed 	| Pubmed* 	| WikiCS 	| Arxiv 	| Products 	|
> |------------------|-----|--------|------|-------|------|-----|--------|
> |       LLMGNN       	| 75.54 	|   69.06  	|  74.98 	|  81.95  	|  66.09 	| 66.14 	|   74.9   	|
> | LLMs-as-Predictors 	| 68.33 	|   66.33  	|  87.33 	|  87.33  	|   71   	| 73.33 	|   75.33  	|
> |         GNN        	| 75.54 	|   69.06  	|  74.98 	|  81.95  	|  66.09 	| 66.14 	|   74.9   	|
>
> The second table compares the number of training samples needed to achieve the performance in the first table. Here, PL stands for pseudo labels generated by LLMs. GT stands for ground truth labels.
> | (Number of   annotations) 	| Cora 	| CiteSeer 	| Pubmed 	| Pubmed* 	| WikiCS 	|  Arxiv 	| Products 	|
> |-------------------------|----|--------|------|-------|------|------|--------|
> |         LLMGNN (PL)        	|  140 	|    120   	|   60   	|   300   	|   200  	|   800  	|    940   	|
> |   LLMs-as-Predictors (PL)  	| 2708 	|   3186   	|  19717 	|  19717  	|  11701 	| 169343 	|  2449029 	|
> |          GNN (GT)         	|  50  	|    50    	|   42   	|   210   	|   40   	|   560  	|    400   	|
>
>
> From these two tables, we can see that **LLMGNN significantly reduces the costs compared to LLMs-as-Predictors**. A recent study [1], shows that the annotations generated by ChatGPT are more than **20 times** cheaper than human annotators. If we use this ratio, we can see that although LLMGNN requires more pseudo labels, the overall costs are still much cheaper than traditional GNN-based pipelines. Using the OGBN-Products dataset as an example, if we estimate the cost, using LLMs-as-Predictors would incur an expense of 1572 dollars, while manual annotation would require at least 20 dollars (assuming each sample has 3 workers on a crowdsourcing platform), and it would also necessitate a considerable amount of waiting time. In contrast, using LLMGNN can complete annotation and prediction in a matter of minutes at a cost of less than one dollar.
>
> [1] Gilardi F, Alizadeh M, Kubli M. Chatgpt outperforms crowd-workers for text-annotation tasks[J]. arXiv preprint arXiv:2303.15056, 2023.

---

> > ### Comment · Reviewer_ugc7 · 2023-11-22
> > **Thanks for the rebuttal**
> >
> > Thank you for addressing all of my concerns with your response. I raise my score accordingly.

---

> > > ### Author Response · Authors · 2023-11-22
> > > **Thanks for your response and support**
> > >
> > > Thanks for your response and support. We are glad to know that our rebuttal has addressed your concerns. Please let us know in case there remain outstanding concerns, and if so, we will be happy to respond.

---

### Meta-Review · Area_Chair_YbPR · 2023-12-07

**Metareview:**

Summary of Scientific Claims: The paper proposes LLM-GNN, a fusion of Large Language Models (LLMs) and Graph Neural Networks (GNNs), aiming to tackle label-free node classification on text-attributed graphs. LLMs excel in zero-shot proficiency but suffer from high inference costs, while GNNs perform well with graph-structured data but require abundant high-quality labels. The method suggests using LLMs for annotation and leveraging GNNs for further predictions, incorporating node selection strategies and confidence-aware annotations to facilitate efficient learning with high-quality annotations.

Strengths: 1. Innovative Approach: The paper introduces an innovative method, LLM-GNN, to address label-free node classification on text-attributed graphs.
2. Methodological Soundness: It balances active node selection, confidence-aware annotations, and post-filtering to ensure annotation quality, diversity, and representativeness.
3. Cost-Effectiveness: Demonstrates cost efficiency, achieving high accuracy on a large dataset (OGBN-PRODUCTS) with annotation costs under 1 dollar.

 Weaknesses and Areas for Improvement:
1. Incremental Performance Gain: Some components like Difficulty-aware active node selection (DA) show only incremental performance gains, raising questions about its effectiveness.
2. Lack of Clarity and Explanation: Explanations about experiments are insufficient, leading to unclear parts within the paper. For instance, clarification on trends in accuracy, confidence score determination, and performance fluctuations in different scenarios.
3. Generalizability and Robustness: The paper's generalizability to other datasets/domains and its robustness to noisy annotations need more exploration and clarification.

The paper's strengths lie in its innovative approach, cost-effectiveness, and a comprehensive methodology. However, it faces challenges in terms of incremental performance improvements, lack of clarity in explanations, and the need for more extensive exploration regarding generalizability and robustness.

**Justification For Why Not Higher Score:**

N/A

**Justification For Why Not Lower Score:**

1. Innovative Approach: The paper introduces a novel method, LLM-GNN, addressing an emerging and challenging problem of label-free node classification on text-attributed graphs. This innovative approach holds promise for further advancements in the field.

2. Significant Contributions: Despite certain weaknesses and incremental gains, the paper offers significant contributions by balancing active node selection, confidence-aware annotations, and post-filtering to ensure annotation quality and efficiency. Its cost-effectiveness, achieving high accuracy at a low annotation cost, is noteworthy.

3. Addressing Limitations: While weaknesses exist, the paper acknowledges these and has attempted to mitigate them. It explores innovative solutions and attempts to balance the limitations of both LLMs and GNNs, which is a notable step toward tackling challenges in this domain.

4. Reviewer Suggestions: The weaknesses highlighted by the reviewers mostly revolve around areas that could be improved or clarified rather than fundamental flaws that render the approach invalid. The paper's potential for improvement based on reviewer suggestions indicates that these weaknesses are not insurmountable obstacles.

In summary, the paper's novelty, significant contributions, attempts to mitigate limitations, and potential for improvement despite identified weaknesses justify its acceptance. The strengths and potential impact of the proposed LLM-GNN approach outweigh its current shortcomings.

---

### Decision · Program_Chairs · 2024-01-16

Accept (poster)